# CAN DIFFUSION MODELS DISENTANGLE?
# A THEORETICAL PERSPECTIVE

## ABSTRACT

This paper introduces a novel theoretical framework to understand how diffusion models can learn disentangled representations under the assumption of an $L^2$ score approximation. We also provide sufficient conditions under which such representations are beneficial for domain adaptation. Our theory offers new insights into how existing diffusion models disentangle latent variables across general distributions and suggests strategies to enhance their disentanglement capabilities. To validate our theory, we perform experiments using both synthetic data generated from latent subspace models and real speech data for non-parallel voice conversion - a canonical disentanglement problem. Across various classification tasks, we found voice conversion-based adaptation methods achieve significant improvements in classification accuracy, demonstrating their effectiveness as domain adaptors. Code will be released upon acceptance.

## 1 INTRODUCTION

Diffusion models (DMs) Sohl-Dickstein et al. (2015); Song & Ermon (2019); Ho et al. (2020) are generative models capable of approximating probability distributions by learning noisy versions of their gradients. While such approaches enjoy both empirical successes (e.g., Ramesh et al. (2022) and theoretical guarantees Chen et al. (2023b); Pabbaraju et al. (2023), they tend to represent the latent structure of the underlying distribution *implicitly*. However, in learning tasks such as controllable generation, it is useful to represent the task-specific latent structure *explicitly* in the generative model to reflect the inductive biases of the problems. One approach, known as *conditional diffusion models* (CDMs), achieves this goal with DMs by labeling such variables and conditioning the model on these labels Wu et al. (2023); Yang et al. (2023); Hudson et al. (2024). However, it remains unclear whether and when CDMs can learn an explicit representation that captures the conditional dependency relations between the variables, especially when some of them are unlabeled. For example, for (approximately) independent latent variables, it is desirable to have a *disentangled* representation with decomposable parts for each variable. Learning such a representation is called *disentanglement*. An intriguing theoretical question then arises: *what are the fundamental limits for CDMs to learn disentangled representations?* A theory capable of answering this question can potentially lead to more powerful, compositional generative models for a wider range of applications.

To answer the question, we focus on one canonical example of the disentanglement problem — *voice conversion* (VC) with *non-parallel* speech recordings. The choice is justified on three grounds. First, the task involves a simple latent variable model but captures the essence of the disentanglement problem. Second, it is practically useful, as many speech data, such as those from underrepresented minorities and subjects with speech impairments, have neither paired target speech nor reliable transcripts for a fully supervised VC. Given only unpaired utterances with speaker identity labels for training, VC tries to change the identity of the source speech to that of the target speech, without modifying other content during inference. To generalize from source-source conversion to source-target conversion, the model has to learn a representation that disentangles the "content" variable from the "speaker" variable in the speech signal during training. Lastly, DM-based VC models (DMVC) have recently revolutionized the field of VC Popov et al. (2022); Choi et al. (2023); Seed Team (2024) and provided new opportunities for generating realistic synthetic data for speech classification tasks. However it is not fully understood how such models perform disentanglement and improve downstream performance of speech classification systems.

The main contribution of this paper is twofold. First, we develop a novel information-theoretic description of the mechanisms behind DM-based disentanglement under a $L^2$ score approximation assumption, and prove the benefits of using multiple imperfectly disentangled representation to improve downstream classification tasks. Further, we validate our theory empirically by conducting classification experiments under domain shift on both synthetic and realistic speech classification datasets.

## 2 DIFFUSION-BASED CONTENT-SPEAKER DISENTANGLEMENT

We will use the intuitive terms of voice conversion to describe the general disentanglement problem. In the content-speaker disentanglement problem, a learner is given a noisy speech signal $X \sim q_\gamma$, which is a function of three random variables $Z$, $G$ and $\Xi$:

$$X := \psi(Z, G, \Xi). \tag{1}$$

The variable $Z \sim q_Z$ is the *content* of the speech that the learner would like to extract for a *downstream classification task* $Y$, such as emotion recognition. Further, $G \sim \gamma$ is the *speaker identity* of the speech, which is *observable* by the learner but contains little information to the downstream task label $Y$. Finally, $\Xi$ is the *hidden noise* independent of $Z$ and $G$ that sets the limit on how well $X$ can predict $Z$ and $G$. More precisely, we make the following independence assumptions.

**Assumption 1.** *The generative process in Equation 1 possesses the following statistical properties:*

1. Disentanglement*: $Z \perp\!\!\!\perp G$;*

2. Conditional disentanglement*: $Y - Z - X - G$ forms a Markov chain;*

3. Bounded predictivity*: $I(Z, G; X) \leq h(X) + \epsilon_\psi - \frac{1}{2}\log(2\pi e)^d \epsilon_\psi$ for some $\epsilon_\psi > 0$.*

The task of *disentanglement* is then to recover $Z$ given $X$ and $G$.

One way to approximate Equation 1, as done in the latest diffusion model-based VC systems, is based on *score matching* Song & Ermon (2019). Given some training speech features $X \sim q_\alpha =: q_{\alpha,0}$ and some *auxiliary variables* $A = a(X)$ for some function $a$ that contains mostly content information, such as the average spectrogram in the DiffVC system Popov et al. (2022), and the speaker feature $\hat{G}$, a diffusion-based voice converter tries to learn $q_\alpha(x)$ by approximating its *gradients* during training by a two-stage process. In the *noising* step, the model injects noise into the input speech following a Markov random process $\{B_t\}_{t \in [0,T]} \sim Q_{[0,T]}$:

$$dX_t = f(X_t^\leftarrow, A, t)dt + \nu(t)dB_t, \ X_0 \sim q_{\alpha,0}, \tag{2}$$

for some *parameter-free* functions $f$ and $\nu$, wehe we will set process for better comparison with prior works Chen et al. (2023b;a). Denote $q_{\alpha,t}$ as the distribution of $X_t$ and $X_t^\leftarrow := X_{T-t}$ and $\hat{G}_t =: g(X_{<t})$ as the noisy speaker embeddings at time $t$, such as those from a speaker verification system. In the *denoising* step, the model learns to recover the clean speech features from the noisy features $X_T$ and the auxiliary variable $A$ by simulating the reverse process:

$$dX_t^\leftarrow = \left(f(X_t^\leftarrow, A, t) + \nu(T-t)^2 \nabla_x \log q_{T-t}(X_t^\leftarrow|A)\right)dt + \nu(T-t)dB_t, \ X_0^\leftarrow \sim q_{\alpha,T}. \tag{3}$$

For the denoising step, the model learns a *score function* $s : [0,T] \times \mathcal{X} \times \mathcal{G} \mapsto \mathbb{R}$ to minimize the score matching objective:

$$L_{\text{match}}(\theta, \phi) := \mathbb{E}_{t,q_{\alpha,t}} \|s_\theta(z_{\phi,t}(X_t), \hat{G}_t, A, t) - \nabla_x \log q_{\alpha,t}(X_t|A)\|^2, \tag{4}$$

where $z_{\phi,t}(X_t) =: \hat{z}_t(X_t) =: \hat{Z}_t$ is some *bottleneck* representation aiming to keep only the content information of $X_t$. While in practice the gradients $\nabla_x \log q_{\alpha,t}$ are not available to the model, the objective can be approximated using *conditional* score matching with $\nabla_x \log q_{\alpha,t|0}(x_t|x_0)$'s:

$$L_{\text{cmatch}}(\theta, \phi) := \mathbb{E}_{t,q_{\alpha,0}} \mathbb{E}_{q_{\alpha,t|0}} \left\| s_\theta(\hat{Z}_t, \hat{G}_t, A, t) + \frac{1}{\sigma^2(t)}(X_t - X_0) \right\|^2 \tag{5}$$

for some time-dependent variance $\sigma(t)$ depending on the noising schedule. In our analysis, we found that another related loss may be needed to reduce bias in the model during inference similar to the

*rectified flow* technique Liu et al. (2023b), where we minimize the score matching objective using generated trajectory:

$$L_{\text{rematch}}(\theta, \phi) := \mathbb{E}_{t, \hat{q}_{\alpha, 0}} \left\| s_\theta(\hat{Z}_t, \hat{G}_t, A, t) - \nabla_x \log q_{\alpha, t|0}(\hat{X}^{\leftarrow}_{T-t}|A) \right\|^2, \quad (6)$$

where the generated trajectory follows the SDE

$$\mathrm{d}\hat{X}^{\leftarrow}_t = \left( f(\hat{X}^{\leftarrow}_t, A, t) + \nu(T-t)^2 s_\theta(\hat{Z}_t, \hat{G}_t, A, t) \right) \mathrm{d}t + \nu(T-t)\mathrm{d}B_t, \ \hat{X}^{\leftarrow}_0 = X_T. \quad (7)$$

The overall training objective of the model is then

$$L(\theta, \phi) = L_{\text{cmatch}}(\theta, \phi) + L_{\text{rematch}}(\theta, \phi). \quad (8)$$

During inference, the VC takes as input the source speech $X^1 \sim q_{\beta,0}$ and the target speech $X^2 \sim q_{\beta,0}$ with a speaker embedding $\hat{G}^2$, and generates converted speech $X^{1\to2} =: X^{2\leftarrow1}_T$ via

$$\mathrm{d}X^1_t = -f(X^1_t, A, t) + \nu(t)\mathrm{d}B_t, \ X_0 \sim q_{\beta,0}, \quad (9)$$

$$\mathrm{d}X^{2\leftarrow1}_t = (f(X^{2\leftarrow1}_t, A, T-t) + \nu(T-t)^2 s_{\hat{\theta}}(t, \hat{z}_t(X^{2\leftarrow1}_t), \hat{G}^2_t, A)dt + \nu(T-t)\mathrm{d}B^{\leftarrow}_t, \ X^{2\leftarrow1}_0 = X^1_T. \quad (10)$$

One intriguing aspect of DMVC is that unlike in the AEVC Qian et al. (2019), there is more than one bottleneck variable involved during inference in Equation 9, namely, the time-dependent $X_T$ and the time-independent $A$. Further, the noise contained in $X_T$ is task-independent and does not fully remove the speaker information in general. Nevertheless, we argue that combining $X_T$ and $A$ indeed constrains the information flow, since the reverse process itself can constrain the information flow due to both the constrained class of score function in each time step. To make the notion more precise, we propose the following definition of *implicit bottleneck*.

**Definition 1.** *For any score function with reverse process $\{X^{\leftarrow}_t\}_{[0,T]}$, the function $\zeta : \mathcal{X}_{[0,t)} \mapsto \mathbb{R}^{d_\zeta}$ is an* implicit bottleneck *at time $t$ if there exists functions $\gamma : \mathcal{G} \times \mathbb{R}^{d_x}_{[0,t]} \mapsto \mathbb{R}^{d_\gamma}$ and $\eta : \mathbb{R}^{d_\zeta} \times \mathbb{R}^{d_\gamma} \mapsto \mathbb{R}^{d_X}$ such that*

$$X^{\leftarrow}_t = \eta(\zeta(X^{\leftarrow}_{<t}), \gamma(G_{<t}, B^{\leftarrow}_{<t})). \quad (11)$$

*Further, define $\zeta^* := \zeta(\hat{X}^{\leftarrow}_{<t^*})$ to be an* implicit bottleneck variable*, where $t^*$ is the largest time step $t$ such that there exists an implicit bottleneck at time $t$.*

One way to design implicit bottleneck is to decompose the score function as

$$s_\theta(\hat{Z}_t, \hat{G}_t, A, t) =: s^Z_\theta(\hat{Z}_t, A, t) + s^G_\theta(\hat{G}_t, t). \quad (12)$$

Plugging this into Equation 7, the reverse SDE yields

$$X^{\leftarrow}_t = \underbrace{X^{\leftarrow}_0 + \int_0^t (f(\hat{X}^{\leftarrow}_\tau, \tau) + \nu(T-\tau)^2 s^Z_\theta(\hat{Z}_\tau, \tau))\mathrm{d}\tau}_{=:\zeta(X^{\leftarrow}_{<t})} + \underbrace{\int_0^t \nu(T-\tau)^2 s^G_\theta(\hat{G}_t, \tau)\mathrm{d}t + \int_0^t \nu(T-\tau)\mathrm{d}B_\tau}_{=:\gamma(G_{<t}, B^{\leftarrow}_{<t})}.$$

Using the implicit bottleneck, the converted speech features generated by the diffusion model-based VC are then

$$\hat{X}^{a\to b} := \hat{\psi}(\zeta(X^a_{<t^*}), \hat{g}(X^b_{<t^*}), \Xi^{a\to b}) =: \hat{\psi}(\zeta^a, \hat{G}^b, \Xi^{a\to b}) \sim \hat{q}_{X^{a\to b}}(\cdot), \ \forall a, b \in \{1, 2\}. \quad (13)$$

for some $\hat{\psi}$ deterministic function and independent random noise $\Xi_{a\to b}$'s introduced by the noising process.

## 2.1 GENERAL CASE

To facilitate analysis, we would need the following definitions and assumptions.

**Definition 2.** *Two variables $X$ and $Y$ are $\epsilon$-disentangled if there exists $\epsilon > 0$ such that $I(X; Y) \leq \epsilon$.*

**Definition 3.** *A VC is $\epsilon$-semantically matched if the target speech $\hat{X}^2$ and the converted speech $\hat{X}^{1\to 2}$ obey $\max_{z\in\mathcal{Z}} d_{\text{TV}}(q_{X^{1\to 2}|Z^1=z}, q_{X^2|Z^2=z}) \leq \epsilon$.*

**Assumption 2.** *The following holds for the diffusion process in Equation 3 and the score matching model trained with Equation 5:*

1. *There exists an implicit bottleneck variable $\zeta^*$ such that*
$$I(\zeta^*; X) \leq I(Z; X) + \epsilon_Z, \tag{14}$$
   *where $t^*$ is the largest time step $t$ such that there exists an implicit bottleneck at time $t$.*

2. *The speaker embedding representation $\hat{G}$ and the true speaker representation $G$ satisfy*
$$\max\{\|\hat{G} - G\|_2, |I(\hat{G}; X) - I(G; X)|\} \leq \epsilon_G \tag{15}$$

3. *The constrained score function is able to approximate the true score function up to an error:*
$$\min_{\theta,\phi} L_{\text{cmatch}}(\theta, \phi) + L_{\text{rematch}}(\theta, \phi) =: L^* \leq \epsilon_{\text{score}}^2. \tag{16}$$

4. *The content variable $Z$ and the speaker variable $G$ are $\epsilon_T$-disentangled given the implicit bottleneck $\zeta^*$:*
$$I(Z; G|\zeta^*) \leq \epsilon_T. \tag{17}$$

5. *Both the content $p_Z$ and speaker distributions $\alpha, \beta$ have bounded support, and the speaker distribution is isotropic.*

6. *The speech feature distribution $q_{\gamma,0}$ has bounded second moment.*

7. *The true score function is Lipschitz in $x$, and the estimated score function $s_\theta(z, g, a, t)$ is $C_{\text{score}}$-Lipschitz in $t$, $z$ and $g$.*

We are now ready to state the main result of this part.

**Theorem 1.** *If Assumption 1-2 hold, and $f(X_t, A, t) = \frac{\nu^2(t)}{2}(\mu(A, t) - X_t)$ for some parameter-free function $\mu$, then $\zeta^*$ and $\hat{G}$ are $\epsilon_D$-disentangled and the diffusion VC is $O(\sqrt{\epsilon_M} + \epsilon_{\text{score}})$-semantically matched for $\epsilon_D = O(\epsilon_\psi + \epsilon_Z + \epsilon_G + \log\frac{(\epsilon_{\text{score}} + \epsilon_G)T}{\epsilon_\psi})$ and $\epsilon_M = O(\sqrt{\epsilon_T + \epsilon_D})$.*

Item 4 in Assumption 2 may seem artificial at first glance, but we provide a failure example in Appendix D to show that it is indeed necessary to preserve the *content* of the source speech. Further, without item 4, we prove a weaker version of Theorem 1 in Appendix C.

## 2.2 SPECIAL CASE: LINEAR SUBSPACE MODEL

One scenario our theory can be applied to is the *latent subspace model* (LSM), previously adopted by classical speaker representation methods such as the *i-vector* model Dehak et al. (2011) and held approximately true for various self-supervised speech representations Liu et al. (2023a).

**Definition 4.** *A latent subspace model is the following generative process:*
$$Z \sim p_Z, \ G \sim \alpha, \ X = A_Z Z + A_G G, \tag{18}$$
*where $A_Z \in \mathbb{R}^{d_x \times d_Z}$, $A_G \in \mathbb{R}^{d_x \times d_G}$ are orthogonal matrices and the subspaces for the content and speaker are orthogonal and span the whole space, i.e., $R(A_Z)^\perp = R(A_G)$ with $d_Z + d_G = d_X$, where $R(A)$ is the column space of matrix $A$. Further, let $X_t$ be the noisy feature variable at time $t$ of the diffusion process and define $Z_t := A_Z^\top X_t$, $G_t := A_G^\top X_t$.*

For LSM, we will prove that the model is able to learn a disentangles representation *without* any auxiliary labels. To this end, we consider the following *regularized* score matching loss with a *decomposable* score function as in Equation 12:
$$L_{\text{match}}(\theta_Z, \theta_G, U, U', V) := \mathbb{E}_{t, q_{\alpha,t}} \|U s_{\theta_Z}(U'^\top X_t) + V s_{\theta_G}(G_t) - \nabla_x \log q_{\alpha,t}(X_t)\|_2^2, \tag{19}$$
$$L_{\text{reg}}(\theta_G, V) := \mathbb{E}_{t, q_{\alpha,t}} \|V s_{\theta_G}(G_t) - \nabla_x \log q_{\alpha,t}(X_t)\|_2^2, \tag{20}$$
$$\tilde{L}_{\text{match}}(\theta_Z, \theta_G, U, \text{proj}_{\mathcal{O}} U, V) := L_{\text{match}}(\theta_Z, \theta_G, U, \text{proj}_{\mathcal{O}} U, V) + \lambda L_{\text{reg}}(\theta_G, V), \tag{21}$$

for some weighting $\lambda > 0$, where $U \in \mathbb{R}^{d_X \times d_U}, V \in \mathbb{R}^{d_X \times d_G}$, $\text{proj}_{\mathcal{M}}$ denotes the projection onto set $\mathcal{M}$ and $\mathcal{O}$ is the Stiefel manifold of size $d_Z$.

Further, we need the following assumption of the subspace score functions.

**Assumption 3.** *The operator norms of the covariance of the content and speaker score functions obey* $\min_{U \in \mathcal{O}} \{\|\mathbb{E}_{t,q_{\alpha,t}} \nabla_{U^\top x} \log p_{U^\top X_t}(U^\top X_t)\nabla_{U^\top x} \log p_{U^\top X_t}(U^\top X_t)^\top\|_{\text{op}}\} =: \lambda_{\min} > 0.$

We show that the objective Equation 21 of LSM recovers the true content and speaker subspaces.

**Theorem 2.** *For the linear subspace model 4 and the objective in Equation 21, and suppose $d_U \leq d_Z$, then any minimizer $(U^*, V^*)$ of Equation 21 satsify $R(U^*) = R(A_Z)$ and $R(V^*) = R(A_G)$.*

**Remark 1.** *Assumption 3 is mild and similar assumption has been made in Chen et al. (2023a).*

**Remark 2.** *$L_{\text{reg}}$ is a novel regularizer that could lead to new disentanglement algorithms with better convergence properties.*

**Remark 3.** *To learn the LSM, one can use an analogous simplified U-net architecture proposed in Chen et al. (2023a), as done in our synthetic experiments.*

Further, we analyze the training dynamics of gradient-based methods for LSM disentanglement by considering the following system of gradient flow equations:

$$\dot{V} = -\nabla_V L_{\text{reg}}(\theta_G^*, V), \tag{22}$$

$$\dot{U} = -\nabla_U L_{\text{match}}(\theta_Z^U, \theta_G^*, U, \text{sg}(\text{proj}_{\mathcal{O}} U), \hat{V}), \tag{23}$$

where $s_{\theta_G^*} = \nabla_g \log \alpha_t(G_t)$, $s_{\theta_Z^U} = \nabla_{U^\top x} \log p_{U^\top X_t}$ and $\dot{x}$ denote the time derivative during the gradient flow, sg denotes the stop-gradient operation, and $\hat{x}$ denotes a stationary point of the gradient flow for $\dot{x}$. The following theorem on the training dynamics requires an additional assumption.

**Assumption 4.** *For nonzero any matrix $U \in \mathcal{O}$ such that $R(U) \cap R(A_Z) \neq \emptyset$,*

$$\|\mathbb{E}_{t,q_{\alpha,t}} \nabla_z \log p_{Z_t}(Z_t)\nabla_{U^\top x} \log p_{U^\top X_t}(U^\top X_t)^\top\|_{\text{op}} > 0.$$

**Theorem 3.** *Suppose $d_U \leq d_Z$, the system of gradient flow equations in Equation 22-23 converges to a stationary point $(\hat{U}, \hat{V})$ such that $R(\hat{U}) = R(A_Z), R(\hat{V}) = R(A_G).$*

**Remark 4.** *Equation 22-23 require access to a score function oracle along subspaces, which can be learned using gradient-based methods up to small error for distributions such as GMM Shah et al. (2023). Analysis with noisy score estimation will be left as future work.*

**Remark 5.** *Once the content subspace $\hat{U}$ is learned using the unconditional score function, a $O(\sqrt{\epsilon_M} + \epsilon_{\text{score}})$-semantically matched VC can be obtained by training another conditional score function $A := \hat{U}^\top X_0$ as auxiliary label, as guaranteed by Theorem 1.*

## 3   DOMAIN ADAPTATION USING IMPERFECTLY DISENTANGLED REPRESENTATIONS

Learning a disentangled representation is especially beneficial for downstream tasks where there is a *domain mismatch* in the speaker variable $G$ during training and testing. This is a common scenario in speech classification tasks such paralinguistic classification, where due to data scarcity, the subjects used during training of the classifier never overlap with those in actual deployment of the classifier. In other words, given speech features $(X_1, Y_1), \cdots, (X_n, Y_n)$ paired with multi-class labels $Y_1, \cdots, Y_n$, where each recording-label pair $(X_i, Y_i)$ is sampled as follows:

1. Sample the content $Z_i \sim q_Z$;
2. Sample the speaker $G_i \sim \alpha$ and noise $\Xi_i$ so that $X_i = \psi(Z_i, G_i, \Xi_i)$;
3. Sample a label $Y_i \sim q_{Y|X=X_i}$.

During inference, the recording-label pairs are sampled from the same type of process but with a different speaker distribution $\beta \neq \alpha$. A multi-class classifier $h : \mathcal{X} \mapsto \{1, \cdots, |\mathcal{Y}|\}$ is then evaluated using the *zero-one loss* defined as:

$$L_P(h) := \mathbb{E}_{(X,Y) \sim P_\alpha} \mathbb{1}[h(X) \neq Y], \tag{24}$$

where $P_\beta(x,y) = q_\beta(x)q_{Y|X=x}(y)$. We are particularly interested in how well the predictor *generalizes* to unknown speaker distribution, or the loss during inference where $X' \sim q_\beta$.

To perform speech classification using voice-converted speech, we propose a simple *adapt-and-vote* scheme. In the adaptation step, we first use a zero-shot VC system to convert the original speech corpus to multiple approximate single-speaker corpora with a random target speaker embedding from some sub-gaussian distribution $\rho$:

$$X^{\gamma,\hat{G}} := \hat{\psi}(\zeta^*(X), \hat{G}, \Xi) \sim q_\gamma^{\hat{G}}, \ X \sim q_\gamma, \ \hat{G} \sim \rho, \ (X^{\gamma,\hat{G}}, Y) \sim P_\gamma^{\hat{G}}. \tag{25}$$

To simplify notations, we will omit the dependence on $\gamma$ when the context is clear. Next, we train a single-speaker, multi-class classifier on the converted training set as $\tilde{f}^G(x) \in \arg\min_{f \in \mathcal{H}} L_{P_\alpha^G}(f)$.

Further, let $\tilde{f}^G(y|x)$ be the *estimated posterior probability* of the model, and let $f^G(y|X) := \mathbb{E}_{X^G}[\tilde{f}^G(y|X^G)|X]$, $f^G(X) = \arg\max_y f^G(y|X)$ be the *conditional expectaction* of $\tilde{f}^G(y|X^G)$ given the original speech.

When the VC system achieves perfect disentanglement, by Theorem 1,

$$d_{\mathrm{TV}}(P_\alpha^G, P_\beta^G) \le \mathbb{E}_{q_{YZ}} d_{\mathrm{TV}}(q_{X^{\alpha,G}|Z}, q_{X^{\beta,G}|Z}) \approx 0, L_{P_\beta^G}(\tilde{f}^G) \approx L_{P_\alpha^G}(\tilde{f}^G) \approx \min_h L_{P_\alpha^G}(h).$$

However, in reality, converted speech from the VC suffers from target speaker-dependent *distortions* and fails to fully close the train-test domain gap. We introduce the notion of *speaker distortion* defined as follows to quantitatively describes this effect.

**Definition 5.** *A random single-speaker classifier $Y^{g'} \sim f^{g'}(\cdot|X)$ is $(\kappa_1, \kappa_2)$-speaker distorted if for all $x \in \mathcal{X}$ and $g \in \mathcal{G}$, $D_{\mathrm{KL}}(f^{g'}(\cdot|x)||f^g(\cdot|x)) \ge \kappa_1 \|g' - g\|_2^{\kappa_2}$.*

To cope with the distortion issue, we propose an additional *majority voting* step using predictions from the single-speaker classifiers. We consider both the *hard* voting scheme and the *soft* voting scheme:

$$f_{\mathrm{mv}}^{\mathrm{hard}}(x) := \arg\max_y \mathbb{E}_{G \sim \rho} \mathbb{1}[f^G(x) = y], \ f_{\mathrm{mv}}^{\mathrm{soft}}(x) := \arg\max_y \mathbb{E}_{G \sim \rho} f^G(y|x). \tag{26}$$

Note that the soft voting scheme is a generalization to the hard voting scheme considered in the theory Theisen et al. (2023), where they simply set $f^G(y|x) = \mathbb{1}[f^G(x) \ne y]$. The soft majority voting scheme is also more widely used in deep ensemble methods based on random initialization Abe et al. (2022). Therefore, we will focus our attention to the soft majority vote case. One can relate the majority vote error rate to the average error rates of *random* classifiers with random predicted label $Y^G \sim f^G(\cdot|X)$. We can also extend the definition of the error rate $L_P(f^G)$ to random classifiers as

$$L_P^{\mathrm{soft}}(f^G) := \mathbb{E}_{(X,Y) \sim P_\beta} \mathbb{E}_{Y^G \sim f^G(\cdot|X)} \mathbb{1}[Y^G \ne Y] = 1 - \mathbb{E}_{(X,Y) \sim P_\beta} f^G(Y|X). \tag{27}$$

By definition we have $L_P^{\mathrm{soft}}(\mathbb{1}[f^G(\cdot) = \cdot]) = L_P(f^G)$.

To evaluate the of ability of $f_{\mathrm{mv}}^{\mathrm{soft}}(x)$ and $f_{\mathrm{mv}}^{\mathrm{hard}}$ to reduce the effect of speaker distortion, we adopt the *ensemble improvement rate* (EIR) Theisen et al. (2023) in Appendix G.

To relate EIR to the amount of speaker distortion, we first make the assumption that the estimated posteriors $f^G(y|x)$'s have bounded magnitudes, as is the case in neural network classifiers with softmax activations.

**Assumption 5.** *For all speech features $x \in \mathcal{X}$, speaker embeddings $g \in \mathcal{G}$ and labels $y \in \mathcal{Y}$, there exists $\delta_1, \delta_2, \delta_3 > 0$ such that the estimated posteriors $f^G(y|x)$ satisfy*

$$\max_{y'} f^G(y'|x) \in \left[\frac{1}{|\mathcal{Y}|} + \delta_1, 1 - \delta_2\right], \ f^G(y|x) \ge \delta_3. \tag{28}$$

Now we provide the proof of upper bounds on EIR for the majority vote classifiers.

**Theorem 4.** *If Assumption 2 and Assumption 5 hold, and the single-speaker posteriors $f^G(Y|X)$ and classifiers $f^G(x)$ are competent (see Appendix G) and $(\kappa_1, \kappa_2)$-speaker distorted, then the EIRs for the hard and soft majority vote classifiers,* $\mathrm{EIR}^{\mathrm{hard}}$ *and* $\mathrm{EIR}^{\mathrm{soft}}$ *satisfy*

$$\mathrm{EIR}^{\mathrm{hard}} \le \frac{3|\mathcal{Y}| - 4}{|\mathcal{Y}|} - \frac{2|\mathcal{Y}| - 2}{|\mathcal{Y}|} \frac{1 - 2\exp(-C'|c\sqrt{d_G} - (\delta^*/\kappa_1)^{1/\kappa_2}|^2)}{\mathbb{E}_\rho L_{P_\beta}(f^G)},$$

$$\mathrm{EIR}^{\mathrm{soft}} \le \frac{4a_1|\mathcal{Y}| - 4a_1 - 1}{|\mathcal{Y}|} + \frac{2|\mathcal{Y}| - 2}{|\mathcal{Y}|} \frac{a_2 + a_3 \exp(-C'|c\sqrt{d_G} - (\delta^*/\kappa_1)^{1/\kappa_2}|^2)}{\mathbb{E}_\rho L_{P_\beta}(f^G)},$$

*where $C', c > 0$ and*

$$\delta^* := (1 - \delta_2) \log \frac{1 - \delta_2}{\frac{1}{|\mathcal{Y}|} + \delta_1} + \left( \frac{|\mathcal{Y}| - 1}{|\mathcal{Y}|} - \delta_1 \right) \log \frac{\frac{|\mathcal{Y}| - 1}{\mathcal{Y}} - \delta_1 - (\mathcal{Y} - 2)\delta_3}{\delta_3},$$

$$a_1 := \frac{1}{|\mathcal{Y}|} + \delta_1 - \delta_3, \ a_2 := 1 + \delta_1, \ a_3 := 2(1 - \delta_2)^2 + 2\delta_2^2 - 4(1/|\mathcal{Y}| + \delta_1)\delta_3.$$

## 4 EXPERIMENT

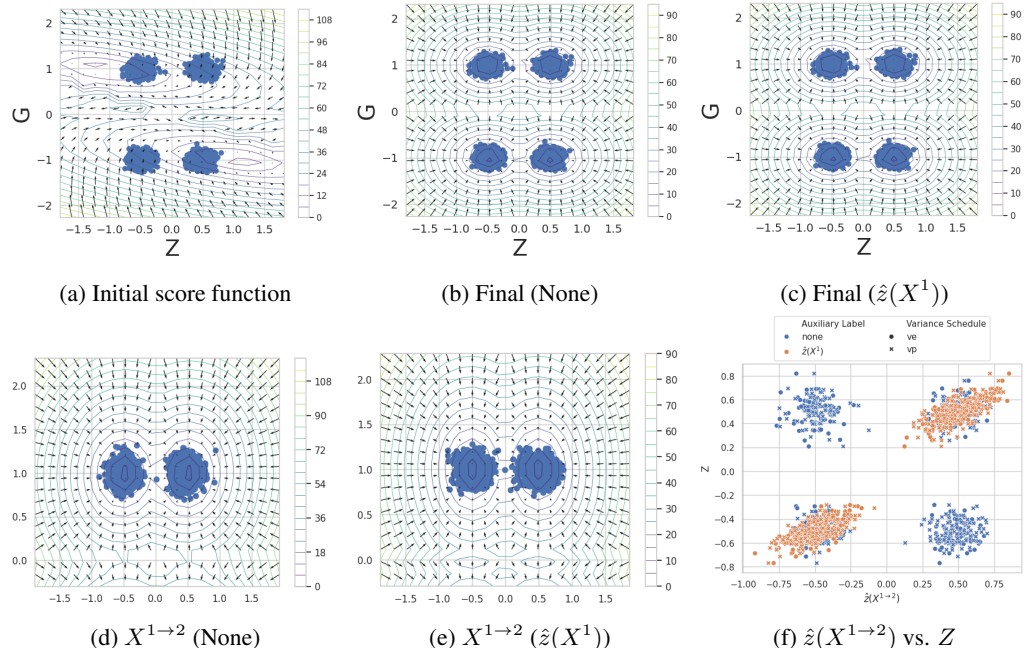

(a) Initial score function      (b) Final (None)      (c) Final $(\hat{z}(X^1))$

(d) $X^{1 \to 2}$ (None)      (e) $X^{1 \to 2} (\hat{z}(X^1))$      (f) $\hat{z}(X^{1 \to 2})$ vs. $Z$

Figure 1: Synthetic disentanglement experiments using 2-d LSGMM with 1-d content ($Z$) and speaker ($G$) subspaces along $x$ and $y$ axes respectively. The gradient fields are computed using the learned unconditional score network $s_{\mu,U}^Z$ and $s_{\nu,V}^G$ and the recovered subspaces learned by both types of score networks.

### 4.1 DISENTANGLEMENT EXPERIMENTS ON SYNTHETIC DATA

To evaluate our theory, we first perform disentanglement experiments on synthetic datasets. To this end, we generate two synthetic dataset using LSGMMs. More details are included in Appendix H.

Table 1: Datasets and VC-adapted classifiers used during realistic data experiments

| | $|\mathcal{Y}|$ | Feature | Classifier | #Classifiers | Reference | VC | DM-based | Reference |
|---|---|---|---|---|---|---|---|---|
| IEMOCAP | 4 | wav2vec 2.0 base | MLP | 8 | Busso et al. (2008) | TriAAN-VC | No | Park et al. (2023) |
| ADReSS | 2 | whisper-medium | SVM | 15 | Luz et al. (2020) | KNN-VC | No | Baas et al. (2023) |
| ALS-TDI | 5 | whisper-medium | SVM | 15 | Vieira et al. (2022) | Diff-VC | Yes | Popov et al. (2022) |

### 4.2 VOICE-CONVERSION ADAPTATION ON REALISTIC DATASETS

To further evaluate our theory, with a particular focus on our theory on VC adaptation, we perform VC adaptation experiments on a variety of realistic datasets with a variety of voice conversion models, as listed in Table 1. The datasets cover diverse speech classification tasks including emotion recognition (IEMOCAP Busso et al. (2008)) and speech biomarker impairment classification such as Alzheimer detection (ADReSS Luz et al. (2020)) and Amyotrophic Lateral Sclerosis (ALS) severity classification (ALS-TDI Vieira et al. (2022)). More details are included in Appendix H.

Table 2: Overall results on realistic datasets. More details can be found in Appendix H. All metrics are between 0-100. A: single (average); B: single (best); MV: majority vote; SMV: soft majority vote.

| VC type | Impairment | | | | | | | | Emotion | | | |
| | ALS-TDI, F1↑ | | | | ADReSS, F1↑ | | | | IEMOCAP, Acc. (5-fold)↑ | | | |
| | A | B | MV | SMV | A | B | MV | SMV | A | B | MV | SMV |
|---|---|---|---|---|---|---|---|---|---|---|---|---|
| No VC | 54.9 | 54.9 | 54.9 | 54.9 | 70.6 | 70.6 | 70.6 | 70.6 | 71.5 | 71.5 | 71.5 | 71.5 |
| Pitch shifting | 55.8 | 60.3 | 57.6 | 61.5 | 71.2 | 77.1 | 77.1 | 68.8 | 60.6 | 55.1 | 61.1 | 61.1 |
| KNN-VC | 55.8 | 61.7 | 64.8 | 49.9 | 71.5 | 79.2 | 79.2 | 83.3 | 70.4 | 69.3 | 71.4 | 71.5 |
| TriAAN-VC | 55.7 | 60.7 | 61.7 | 53.3 | 72.4 | 75.0 | 77.1 | 83.3 | 65.1 | 64.1 | 66.8 | 67.2 |
| Diff-VC | 47.0 | 51.2 | 50.3 | 49.2 | 65.6 | 69.4 | 66.7 | 70.8 | 87.0 | 94.3 | 96.5 | 97.2 |

## 4.3 RESULTS ON SYNTHETIC DATASETS

We conduct experiments on *latent subspace GMM*s (LSGMM), which are LSM with each subspace being a Gaussian mixture models (GMM). First, we visualize the process of disentanglement of DM for the 2-D LSGMM with 1-D content and speaker subspaces by plotting the gradient fields learned by the unconditional score function $s_{\mu,U}^Z$ and $s_{\nu,V}^G$ and the recovered subspaces learned by both the conditional and unconditional models, as shown in Figure 1.

Both the unconditional and conditional score networks are able to disentangle $Z$ and $G$ by approximating the correct subspaces and the corresponding content and speaker score functions as shown in Figure 1b and Figure 1c respectively. Further, Figure 1d and Figure 1e demonstrate that both models are able to approximate the target speech distribution $X^{2\to 2}$ using the converted speech $X^{1\to 2} \sim q_{X^{1\to 2}}$ from the mismatched content-speaker pair $(\hat{Z}^1, \hat{G}^2)$, as predicted by Theorem 6. However, as shown in Figure 1f, while the converted speech by the unconditional model has content variables evenly distributed across different mixtures, the content variable of the conditional model is strongly correlated with the source speech, showing that the conditional model is able to preserve semantic information while the unconditional model is not. This suggests at least the standard noise schedule makes $X_T$ alone a poor information bottleneck for VC purposes, and good auxiliary labels can be essential for learning semantically matched VCs as predicted by Theorem 1.

Further, Figure 2 shows the subspace recovery error as a function of the number of columns of the learnable matrix $U$. As predicted by Theorem 2, the LSGMM achieves the smallest subspace reconstruction error when the dimension of $U$ matches the true content subspace at $d_{\hat{Z}} = 5$, and the result is consistent across different variance schedules. Further, as all the score networks are neural networks trained using gradient-based method, the result also provides empirical support for Theorem 3. Also, we found that the conditional and unconditional models achieve similar level of error, suggesting that the conditional model training is more effective for learning the semantic correspondence between the source and target speech than refining the subspace.

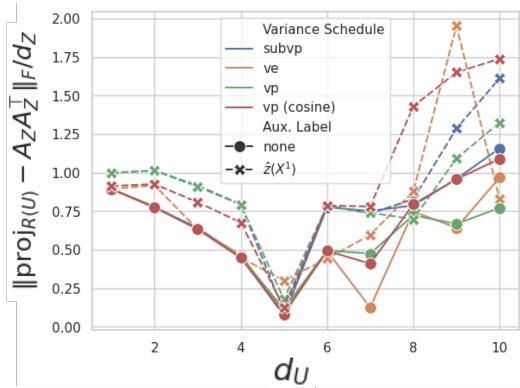

Figure 2: Subspace recovery error vs. learnable subspace dimension for an LSGMM using various variance schedules and two types of auxiliary labels. The score function is a small multilayer perceptron (MLP) described in Appendix H. The total dimension $d_X = 10$, and the true content dimension subspace $d_Z = 5$. The subspace recovery error is the distance between the projection matrix of two spaces normalized by $d_Z$, and takes value between $[0, 2]$. DM consistently recovers the correct content subspace and achieves disentanglement when the learnable subspace dimension $d_U$ matches $d_Z$.

## 5 RESULTS ON REALISTIC DATASETS

The results on the realistic datasets are summarized in Table 2 and Figure 3.

Each plot depicts the classification performance as a function of the number of target speakers used to perform VC adaptation. For each target speaker number, we randomly select 4 speaker combinations.

**Adding target speakers reduces speaker distortion**   As shown in Figure 3, macro-F1 improves steadily as the number of speakers increases, suggesting that having more target speakers can reduce the effect of speaker distortion as predicted by Theorem 4. The trend is noisier for speech impairment detection datasets such as ALS-TDI and ADReSS, which makes sense as they are relatively small in size.

**Different VC excels at different tasks**   However, we found that different VCs excel at different tasks.

For ALS severity classification as shown in Table 2, KNN-VC achieves the best performance among the VCs, reaching 65% macro-F1 with 15 target speakers and hard majority voting, compared to 54.9% when training without VC adaptation and 61.7% with pitch shifting. For cognitive impairment detection as shown in Table 2, TriAAN-VC performed the best followed by the KNN-VC method, both achieved 83.3% macro-F1 with soft majority voting, which is 12.7% better than methods without VC adaptation and 14.5% and 6.2% better than the pitch shifting adaptation using hard and soft majority voting respectively. On IEMOCAP, we found that Diff-VC performs the best, reaching an average of 97.2% accuracy, which is 25.7% better than the no-VC classifier and 36.1% than the pitch shifting adaptation. Though a phenomenon out of the scope of predictions by our theory, we hypothesized that such "specialization" of the VC methods is due to the different level of generalization ability of different VCs to latent variables *other than* the speaker identity, such as recording conditions and health conditions of the speaker. For instance, Diff-

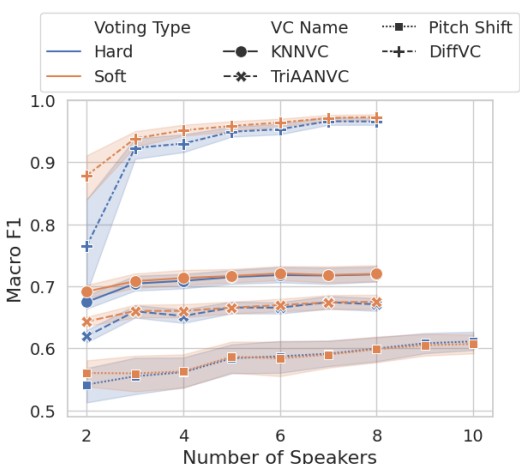

Figure 3: A closer look into classification performance vs. number of target speakers for VC adapation on IEMOCAP. Having more target speakers for conversion consistently improves classification results.

VC does not perform well on ALS compared to KNN-VC, probably due to the domain mismatch between the health conditions of its training set, which contains little pathological speech, compared to KNN-VC which uses the WavLM representation trained on much larger speech dataset with diverse speech.

**Tradeoff between classifier accuracy and diversity**   As to the advantage of hard vs. soft voting, we observe different trends across different datasets and VC methods. On ALS-TDI, hard voting works better than soft voting by 8.4% and 16% for the best two methods Diff-VC and KNN-VC, though worse by 3.9% and 1.3% for pitch shifting and Diff-VC. On IEMOCAP, the gap between soft and hard voting is negligible, with soft majority voting shows a 0.1%-0.7% edge over hard majority voting across VC methods. On ADReSS, we found soft voting methods to be better than hard voting for all the VC methods by $4.1\% - 6.2\%$, while worse for the pitch shifting method by 8.3% (68.8% vs. 77.1%). Since soft voting uses a random classifier for voting, it tends to perform well when the model is "confidently" correct and "hesitantly" wrong, as it puts more weights on confident classifiers than hesitant ones. This suggests that the average confidence score estimated in terms of the classifier posteriors on incorrect examples will be high for classifier ensembles that perform well with hard voting than soft voting.

## 6 RELATED WORKS

**Disentangled representation learning**  The concept of disentanglement we adopt is first defined explicitly as a generalization of statitical independence Tishby et al. (1999) based on mutual information, though other definitions exist, e.g., Higgins et al. (2018). Disentanglement is a crucial concept for deep learning in fields such as representation learning Bengio et al. (2013); Schmidhuber (1992); Tschannen et al. (2018) and voice conversion Qian et al. (2019); Wang et al. (2021a); Popov et al. (2022), and neural network-based architectures have been proposed to learn disentangled representation Hsu et al. (2017); Higgins et al. (2017); Kim & Mnih (2018); Chen et al. (2016); Wu et al. (2023); Yang et al. (2023); Hudson et al. (2024) among others, though theoretical understanding of such models remain limited. To understand the learnability of disentangled representation, Locatello et al. (2019) proved a no-free-lunch theorem on disentanglement inspired by classical results in independent component analysis Comon (1994). Motivated by the task of VC, Qian et al. (2019) proves that for the content-speaker latent variable model, content-speaker disentanglement is indeed possible when the speaker variable is observed, a result our theory extends to DMVCs and generalizes to noisy, continuous content and speaker variables.

**Diffusion model theory**  Early works on DMs focus on their ability to learn general data distributions, under different assumptions on statistical properties of the data distribution such as log-Sobelev inequality Lee et al. (2022), and bounded moments Block et al. (2020); Chen et al. (2023b) and score approximation accuracy in terms of $L^\infty$-accuracy Bortoli et al. (2021) and $L^2$-accuracy Lee et al. (2022); Chen et al. (2023b). Others attempt to understand the benefit of DM over maximum-likelihood-based generative models Pabbaraju et al. (2023). More recent works have started to analyze the ability of DM to learn latent low-dimensional subspace Chen et al. (2023a) and manifold structure Bortoli et al. (2022). Further, Fu et al. (2024) studies the convergence properties of CDMs for a variety of latent variable learning tasks and the role of classifier-free guidance in such tasks.

**Ensembling theory**  Our theory on combining multiple imperfectly disentangled representation for domain adaptation is inspired by earlier works on the statistical learning theory of ensembling methods Langford & Shawe-Taylor (2002); Germain et al. (2015); Masegosa et al. (2020); Theisen et al. (2023), which has seen success in both machine learning Breiman (1996; 2001) and deep learning applications, e.g., Ovadia et al. (2019); Fort et al. (2019); Ashukha et al. (2020). Along this direction, Langford & Shawe-Taylor (2002) gives a simple PAC-Bayes bound on the error rate of majority vote classifier to be no more than twice of the average error of the individual classifiers. Germain et al. (2015) proposed the $\mathcal{C}$-bounds for binary majority vote classifiers in terms of their average pairwise disagreement, which could be much tighter than the simple bound. Masegosa et al. (2020) extends the $\mathcal{C}$-bound to general multi-class setting and demonstrate that it is strictly better than the average single-classifier errors under stronger conditions. Theisen et al. (2023) relaxes the condition in Masegosa et al. (2020) and improve their bound by a factor of 2. Others have used different loss functions such as cross entropy Abe et al. (2022) and challenge the connection between the diversity of classifiers and the success of ensemble methods.

## 7 CONCLUSION

In this work, we propose a theory for understanding the ability of diffusion model to disentangle latent variables and how imperfect disentanglement in general can benefit classification tasks. By studying the roles of diffusion noise, auxiliary variables, score network design and training dynamics, our theory provides a unified framework for DM-based disentanglement. Rigorous synthetic experiments as well as extensive experiments on realistic datasets provide evidence to support our theory. Our experiment also demonstrates the limitations of current DMVC models, such as the robustness against certain domain shifts not prevented by disentanglement during training. Future works include thoroughly understanding the training dynamics of the DM-based disentanglement and apply our theory to design more powerful DMVCs.

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

APPENDIX

CONTENTS

## A  PROOF OF THEOREM 1

To prove Theorem 1, we first prove the following theorem.

**Theorem 5.** *Suppose Assumption 1-2, the following holds*

$$
\max\{I(\zeta^*; \hat{G}), I(\zeta^*; G)\} \leq \max\{\epsilon_\psi + \epsilon_Z + \epsilon_G + \frac{1}{2}\log\frac{2TC_1\epsilon_{\text{score}}^2}{\epsilon_\psi^2},
$$

$$
\epsilon_\psi + \epsilon_Z + \frac{1}{2}\log\frac{2TC_1\left(\epsilon_{\text{score}}^2 + 2\sqrt{2}C_{\text{score}}\epsilon_G\epsilon_{\text{score}} + C_{\text{score}}^2\epsilon_G^2 T\right)}{\epsilon_\psi^2}\} =: \epsilon_D. \quad (29)
$$

Provided that Theorem 5 is true, we can then prove Theorem 1 as follows.

First, by Assumption 1, Equation 17 and Theorem 5,

$$
I(Z, \zeta^*; G) = I(\zeta^*; G|Z) = I(Z; G|\zeta^*) + I(\zeta^*; G) \leq \epsilon_T + \epsilon_D.
$$

Therefore, by Pinsker's inequality and the fact that $p_Z, \alpha$ both have bounded support by Assumption 2,

$$\max_{z \in \mathcal{Z}, g \in \mathcal{G}} d_{\text{TV}}(p_{\zeta^*|z,g}, p_{\zeta^*|z}) \leq \max_{z \in \mathcal{Z}, g \in \mathcal{G}} \sqrt{\frac{1}{2} D_{\text{KL}}(p_{\zeta^*|z,g} || p_{\zeta^*|z})}$$

$$\leq \frac{\sqrt{\frac{1}{2} I(\zeta^*; G|Z)}}{\min_{z' \in \mathcal{Z}, g' \in \mathcal{G}} p_Z(z') \alpha(g')}$$

$$\leq \frac{\sqrt{\epsilon_T + \epsilon_D}}{\sqrt{2} \min_{z' \in \mathcal{Z}, g' \in \mathcal{G}} p_Z(z') \alpha(g')} =: \epsilon_M.$$

Fixing $Z^1 = Z^2 = z, G^1 = g^1, G^2 = g^2$, and apply data processing inequality for $d_{\text{TV}}$ and Pinsker's inequality again,

$$d_{\text{TV}} \left( q_{X^{1 \rightarrow 2}|z,g^1,g^2}(x), q_{X^{2 \rightarrow 2}|z,g^2}(x) \right)$$

$$= d_{\text{TV}} \left( \int p_{\zeta^1|z,g_1}(\hat{z}) p_{X^{1 \rightarrow 2}|\hat{z},g_2}(x) \mathrm{d}\hat{z}, \int p_{\zeta^2|z,g_2}(\hat{z}') p_{X^{2 \rightarrow 2}|\hat{z}',g_2}(x) \mathrm{d}\hat{z}' \right)$$

$$\leq d_{\text{TV}} \left( p_{\zeta^1|z,g_1}, p_{\zeta^2|z,g_2} \right) \leq d_{\text{TV}} \left( p_{\zeta^1|z,g_1}, p_{\zeta^1|z} \right) + d_{\text{TV}} \left( p_{\zeta^2|z,g_2}, p_{\zeta^2|z} \right)$$

$$\leq 2 \frac{\sqrt{D_{\text{KL}}(p_{\zeta^*|Z,G} || p_{\zeta^*|Z})/2}}{\min_{z,g} p_Z(z) \alpha(g)} \leq \frac{\sqrt{2\epsilon_M}}{\min_{z,g \in \mathcal{Z} \times \mathcal{G}} p_Z(z) \alpha(g)}.$$

Marginalizing over $g^1, g^2$ and use Jensen's inequality and triangle inequality yields

$$d_{\text{TV}} \left( q_{X^{1 \rightarrow 2}|Z^1=z}, q_{X^2|Z=z} \right) \leq d_{\text{TV}} \left( q_{X^{1 \rightarrow 2}|Z^1=z}, q_{X^{2 \rightarrow 2}|Z^2=z} \right) + d_{\text{TV}} \left( q_{X^{2 \rightarrow 2}|Z^1=z}, q_{X^2|Z^2=z} \right)$$

$$\leq \frac{\sqrt{\epsilon_M}}{\sqrt{2} \min_{z,g \in \mathcal{Z} \times \mathcal{G}} p_Z(z) \alpha(g)} + \frac{\epsilon_{\text{score}}}{\sqrt{2} \min_{z \in \mathcal{Z}} p_Z(z)}.$$

## B  PROOF OF THEOREM 5

We will need the following lemma to lower bound $I(\zeta, \hat{G}; X)$.

**Lemma 1.** *Given Assumption 1-2, the following inequalities hold for the VC system trained on Equation 8:*

$$I(\zeta^*, \hat{G}; X) \geq h(X) - \frac{1}{2} \log(2\pi e)^{d_X} 2TC_1 \epsilon_{\text{score}}^2, \tag{30}$$

*for some $C_1 > 0$.*

### B.1  PROOF OF LEMMA 1

In this proof, we omit the dependence of distribution on $\alpha$ let $d_X = d$ and $\nu(t) \equiv \sqrt{2}$ for notational and analytical simplicity, and Equation 3 becomes:

$$\mathrm{d}X_t^{\leftarrow} = (X_t^{\leftarrow} - \mu(A, t) + 2\nabla_x \log q_{T-t}(X_t^{\leftarrow}|A))\mathrm{d}t + \sqrt{2}\mathrm{d}B_t^{\leftarrow}. \tag{31}$$

Define the perturbed version of $X_\epsilon \sim q_{\epsilon|0} = \mathcal{N}((1 - \epsilon')x, \epsilon^2 I_d)$ for $1 - \epsilon' := \sqrt{1 - \epsilon^2}$. By the true distribution $q_0$ is sub-gaussian with Lipschitz score function, as guaranteed by item 5 and 6 of Assumption 2, we can show that the drift term

$$\Delta_t := \sqrt{2}(s_\theta(\hat{Z}_t, \hat{G}_t, A, t) - \nabla_x \log q_{T-t|\epsilon}(X_t^{\leftarrow}))$$

satisfies the Novikov's condition using an analysis similar to Lemma 11 and 13 of Chen et al. (2023a) for the undiscretized and discretized cases respectively. Therefore, we can apply Girsanov's theorem Chen et al. (2023b) to the SDE in Equation 3 under a different measure $P_{[0,T]}$ defined as

$$B_t \sim P_t = Q_t \exp \left( \int_0^t \langle \Delta_\tau, \mathrm{d}B_\tau \rangle - \frac{1}{2} \int_0^t \|\Delta_\tau\|^2 \mathrm{d}\tau \right), \tag{32}$$

is the same as the following SDE

$$\mathrm{d}\hat{X}_t^{\leftarrow} = (\hat{X}_t^{\leftarrow} - \mu(A, t) + 2s_\theta(\hat{Z}_t^{\leftarrow}, \hat{G}_t, A, t))\mathrm{d}t + \sqrt{2}\mathrm{d}\beta_t, \hat{X}_0^{\leftarrow} = X_T, \tag{33}$$

where $\beta_t := B_t^{\leftarrow} - \int_0^t \Delta_\tau \mathrm{d}\tau$'s form a Brownian motion. Let the distribution of $\hat{X}_t | X_0 = \hat{X}_{T-t}^{\leftarrow} | X_0$ be $\hat{q}_{t|0}$ and the distribution of the whole process be $\hat{Q}_{[\epsilon, T]|0}$, then the theorem suggests

$$D_{\mathrm{KL}}(q_{t|0} || \hat{q}_{t|0}) = \int_{T-\epsilon}^{t-\epsilon} \mathbb{E}_{q_{\tau|0}} \left[ \|\Delta_\tau\|^2 \,\middle|\, X_0 = x \right]. \tag{34}$$

Further, by the property of the O-U process,

$$\tilde{X}_\epsilon := \tilde{X}_{T-\epsilon}^{\leftarrow} := \hat{X}_{T-\epsilon}^{\leftarrow} - \int_0^{T-\epsilon} e^{-(T-\epsilon-\tau)} \Delta_\tau \mathrm{d}\tau \sim q_{\epsilon|0}. \tag{35}$$

Now, by Assumption 2,

$$\mathbb{E}_{\hat{Q}_{[0,T]|0}} \left[ \left\| \hat{X}_{T-\epsilon}^{\leftarrow} - x \right\|^2 \,\middle|\, X_0 = x \right]$$

$$= \mathbb{E}_{\hat{Q}_{[0,T]|0}} \left[ \left\| \hat{X}_{T-\epsilon}^{\leftarrow} - x - \int_0^{T-\epsilon} e^{-(T-\epsilon-\tau)} \Delta_\tau \mathrm{d}\tau + \int_0^{T-\epsilon} e^{-(T-\epsilon-\tau)} \Delta_\tau \mathrm{d}\tau \right\|^2 \,\middle|\, X_0 = x \right]$$

$$\leq 2 \left( \mathbb{E}_{q_{\epsilon|0}} \left[ \|\tilde{X}_\epsilon - x\|^2 \,\middle|\, X_0 = x \right] + \mathbb{E}_{\hat{Q}_{[0,T]|0}} \left[ \left\| \int_0^{T-\epsilon} e^{-(T-\epsilon-\tau)} \Delta_\tau \mathrm{d}\tau \right\|^2 \,\middle|\, X_0 = x \right] \right)$$

$$\leq 2\mathbb{E}_{\hat{Q}_{[0,T]|0}} \left[ \left( \int_0^{T-\epsilon} \|\Delta_\tau\| \,\mathrm{d}\tau \right)^2 \,\middle|\, X_0 = x \right] + 2\epsilon^2$$

$$\leq 2(T-\epsilon) \int_0^{T-\epsilon} \mathbb{E}_{\hat{q}_{\tau|0}} \left[ \|\Delta_\tau\|^2 \,\middle|\, X_0 = x \right] \mathrm{d}\tau + 2\epsilon^2 + 2\epsilon'^2 \|x\|^2, \tag{36}$$

where the first inequality uses the inequality $(x + y)^2 \leq 2(x^2 + y^2)$, and the second inequality uses the triangle inequality, and the last inequality uses Cauchy's inequality.

Marginalizing Equation 36 over $q_0$ and applying Equation 16 yields

$$\mathbb{E}_{q_0} \int_0^{T-\epsilon} \mathbb{E}_{\hat{q}_{\tau|0}} \left[ \|\Delta_\tau\|^2 \,\middle|\, X_0 \right] \mathrm{d}\tau = \int_0^{T-\epsilon} \mathbb{E}_{\hat{q}_\tau} \|\Delta_\tau\|^2 \,\mathrm{d}\tau \leq \epsilon_{\mathrm{score}}^2.$$

Moreover, use the fact that $q_0$ has bounded second moment,

$$\mathbb{E}_{\hat{Q}_{[0,T]|0}} \left\| \hat{X}_{T-\epsilon}^{\leftarrow} - X_0 \right\|^2 \leq 2\epsilon^2 + 2\epsilon'^2 C_0 + 2(T-\epsilon)\epsilon_{\mathrm{score}}^2 \leq 2TC_1\epsilon_{\mathrm{score}}^2, \tag{37}$$

by choosing $\max\{\epsilon, \epsilon'\} < \epsilon_{\mathrm{score}}$ for some $C_0, C_1 > 0$.

To proceed, using the maximum entropy inequality:

$$h(\hat{X}_T^{\leftarrow} | X_0) \leq \lim_{\epsilon \to 0} \frac{1}{2} \log 2\pi e \mathbb{E} \left[ \|\hat{X}_{T-\epsilon}^{\leftarrow} - X_\epsilon\|^2 \right]$$

$$= \frac{1}{2} \log(2\pi e)^d \lim_{\epsilon \to 0} \mathbb{E}_{q_0} \mathbb{E}_{t, \hat{q}_{t|0}} \left[ \|\hat{X}_{T-\epsilon}^{\leftarrow} - X_\epsilon\|^2 | X_0 \right] \mathrm{d}x = \frac{1}{2} \log(2\pi e)^d 2TC_1\epsilon_{\mathrm{score}}^2. \tag{38}$$

Lastly, by the data processing inequality:

$$I(\zeta^*, \hat{G}; X_0) \geq I(\hat{X}_T^{\leftarrow}; X_0) \geq h(X_0) - \frac{1}{2} \log(2\pi e)^d 2TC_1\epsilon_{\mathrm{score}}^2. \tag{39}$$

## B.2 MAIN PROOF

Using Lemma 1, we are able to prove that the noisy content variable $\hat{Z}_t$ and the speaker identity $G$ are approximately disentangled. By definition, the conditional independence relations $\zeta^* - X_0 - \hat{G}$

and $\zeta^* - X_0 - G$ hold and

$$
\begin{aligned}
I(\zeta^*, \hat{G}; X) &= h(\zeta^*, \hat{G}) - h(\zeta^*, \hat{G}|X) \\
&= h(\zeta^*, \hat{G}) - h(\zeta^*|X) - h(\zeta^*|X) \\
&= I(\zeta^*; X) + I(\hat{G}; X) - I(\zeta^*; \hat{G}) \\
&\le I(Z; X) + I(G; X) - I(\zeta^*; \hat{G}) + \epsilon_Z + \epsilon_G \\
&= I(Z, G; X) - I(\zeta^*; \hat{G}) + \epsilon_Z + \epsilon_G,
\end{aligned}
$$

where the last inequality uses Assumption 2. Further, by Lemma 1 and Assumption 1 (3),

$$
\begin{aligned}
I(\zeta^*; \hat{G}) &\le I(Z, G; X) - I(\zeta^*, \hat{G}; X) + \epsilon_Z + \epsilon_G \\
&\le h(X) + \epsilon_\psi - \frac{1}{2}\log(2\pi e)^d \epsilon_\psi^2 - \left( h(X) - \frac{1}{2}\log(2\pi e)^d 2TC_1\epsilon_{\text{score}}^2 \right) + \epsilon_Z + \epsilon_G \\
&= \epsilon_\psi + \epsilon_Z + \epsilon_G + \frac{1}{2}\log\frac{2TC_1\epsilon_{\text{score}}^2}{\epsilon_\psi^2}.
\end{aligned}
$$

Similarly,

$$
I(\zeta^*, G; X) = I(\zeta^*; X) + I(G; X) - I(\zeta^*; G) \le I(Z, G; X) - I(\zeta^*; \hat{G}) + \epsilon_Z,
$$

Let $\Delta_t = \|s_{\theta^*}(\hat{Z}_t^*, \hat{G}, t) - \nabla_x \log q_{\alpha, t|0}(X_t|X_0)\|_2$, and from Assumption 2 (Equation 15 and 16),

$$
\begin{aligned}
&\|s_{\theta^*}(\hat{Z}_t^*, G, A, t) - \nabla_x \log q_{\alpha, t|0}(X_t|X_0)\|_2^2 \\
&\le \left( \|s_{\theta^*}(\hat{Z}_t^*, \hat{G}, t) - \nabla_x \log q_{\alpha, t|0}(X_t|X_0)\|_2 + C_{\text{score}}\|\hat{G} - G\|_2 \right)^2 = (\Delta_t + C_{\text{score}}\epsilon_G)^2
\end{aligned}
$$

As a result,

$$
\begin{aligned}
&\mathbb{E}_{t, q_{\alpha, 0}}\mathbb{E}_{q_{\alpha, t|0}}\|s_{\theta^*}(\hat{Z}_t^*, G, t) - \nabla_x \log q_{\alpha, t|0}(X_t|X_0)\|^2 \\
&\le \mathbb{E}_{t, q_{\alpha, 0}}\mathbb{E}_{q_{\alpha, t|0}}[\Delta_t^2 + 2C_{\text{score}}\epsilon_G\Delta_t + C_{\text{score}}^2\epsilon_G^2] \\
&\le L^* + 2C_{\text{score}}\epsilon_G\sqrt{L^*} + C_{\text{score}}T\epsilon_G^2 \le \epsilon_{\text{score}}^2 + 2C_{\text{score}}\epsilon_G\epsilon_{\text{score}} + C_{\text{score}}^2\epsilon_G^2 T, \quad\quad (40)
\end{aligned}
$$

where we use the Lipschitz property of $s_\theta$ on $\hat{G}$ in item 5, Assumption 2. Then applying Lemma 1 on $\hat{Z}, G$ and Equation 40 in place of $Z, G$ and Assumption 1 (3) yields the desired bound on $I(\zeta^*; G)$.

## C  PROOF OF A WEAKER VERSION OF THEOREM 1 WITHOUT ITEM 4 OF ASSUMPTION 2

**Theorem 6.** *Under Assumption 1-2 except Assumption 2.4, the target speaker distribution $\hat{q}_{X^2}$ and the converted speaker distribution $\hat{q}_{X^1 \to 2}$ satisfy*

$$
d_{\text{TV}}(q_{X^1 \to 2}, q_{X^2}) \le \frac{\sqrt{2\epsilon_D}}{\min_g \alpha(g)} + \frac{1}{\sqrt{2}}\epsilon_{\text{score}}. \quad\quad (41)
$$

By data processing inequality,

$$
\begin{aligned}
&d_{\text{TV}}\left(q_{X^2 \to 2}, q_{X^1 \to 2}\right) \\
&= d_{\text{TV}}\left( \int p_{\zeta^1|g^1}(\hat{z})p_{\hat{X}_T^\leftarrow|\hat{z}, g^2}(x)\mathrm{d}\hat{z}, \int p_{\zeta^1|g^2}(\hat{z})p_{\hat{X}_T^\leftarrow|\hat{z}, g^2}(x)\mathrm{d}\hat{z} \right) \\
&\le d_{\text{TV}}\left( p_{\zeta^1|g^1}, p_{\zeta^1|g^2} \right) \le d_{\text{TV}}\left( p_{\zeta^1|g^1}, p_{\zeta^1} \right) + d_{\text{TV}}\left( p_{\zeta^1|g^2}, p_{\zeta^1} \right) \\
&\le 2\frac{\sqrt{D_{\text{KL}}(p_{\zeta^* G}||p_{\zeta^* \alpha})/2}}{\min_g \alpha(g)} = \frac{\sqrt{2\epsilon_D}}{\min_g \alpha(g)}. \quad\quad (42)
\end{aligned}
$$

where the second to last equality uses the fact that $\zeta^1, G^1$ has identical distribution as $\zeta^2, G^2$ and $X^{1\to 2}|\zeta^1, G^2$ and $X^{2\to 2}|\zeta^2, G^2$ are identically distributed.

Lastly, by applying Assumption 1 and Girsanov's theorem Chen et al. (2023b),

$$d_{\mathrm{TV}}(q_{X^2}, q_{X^2 \to 2}) \leq \epsilon_{\mathrm{score}}/\sqrt{2}. \tag{43}$$

Combining Equation 42 and Equation 43 and using triangle inequality:

$$d_{\mathrm{TV}}(\hat{q}_{X^1 \to 2}, \hat{q}_{X^2}) \leq d_{\mathrm{TV}}(\hat{q}_{X^1 \to 2}, \hat{q}_{X^2 \to 2}) + d_{\mathrm{TV}}(\hat{q}_{X^2 \to 2}, \hat{q}_{X^2}) \leq \frac{\sqrt{2\epsilon_D}}{\min_g \alpha(g)} + \frac{1}{\sqrt{2}}\epsilon_{\mathrm{score}}.$$

## D    FAILURE EXAMPLE

Consider the following example.

**Example 1.** *Under the same independence relations in Assumption 1, let $X := [X(1), X(2)] = [Z + \Xi(1), G + \Xi(2)]$ with $Z \sim \mathcal{N}(0,1)$, $G \sim \mathrm{Unif}\{-1,1\}$, $\Xi(1), \Xi(2) \sim \mathcal{N}(0,\epsilon)$. Further, let the inputs to the score function be $\hat{G} = X(2) + \Xi_G$, $\hat{Z}_t = X_t(1)\,\mathrm{sign}(\hat{G})$, $\Xi_G \sim \mathcal{N}(0,\epsilon)$, and we let the noising schedule $\sigma(t)$'s unspecified[1]. Then it can be shown that $\hat{G}$ and $\hat{Z}_t$ satisfy all conditions except 4 in Assumption 2 but $I(\hat{Z}_T; \hat{G}|Z) \xrightarrow{\epsilon, \sigma(T) \to 0} \infty$.*

Then it can be shown that

$$\mathrm{sign}(\hat{G}^a) \sim \mathrm{Unif}\{-1,1\}, \hat{Z}_t^a \sim \mathcal{N}(0, 1 + \epsilon^2 + \sigma(t)^2), \hat{Z}^a|\hat{G}^a \sim \mathcal{N}(0, 1 + \epsilon^2 + \sigma(t)^2),$$

and that $\zeta^* = \hat{Z}_T$ and $\hat{G}$ are independent:

$$I(\hat{Z}_T; \hat{G}) = h(\hat{Z}_T) - h(\hat{Z}_T|\hat{G}) = \frac{1}{2}\log 2\pi e(1 + \epsilon^2 + \sigma(T)^2) - \frac{1}{2}\log 2\pi e(1 + \epsilon^2 + \sigma(T)^2) = 0$$

Further, $\hat{Z}_T$ and $\hat{G}$ can be proved to contain most information of $X$:

$$I(\hat{Z}_T, \hat{G}; X) = h(X) - h(X|\hat{Z}_T, \hat{G}) = h(X(1)) + h(X(2)) - h(X|\hat{Z}_T, \hat{G})$$

$$\geq \frac{1}{2}\log 2\pi e(1 + \epsilon^2) + \frac{1}{2}\log 2\pi e\epsilon^2 - h(X_0(1)|X_T(1)) - h(X_0(2)|\hat{G})$$

$$= \frac{1}{2}\log 2\pi e(1 + \epsilon^2)\epsilon^2 - \frac{1}{2}\log 2\pi e / \left(\frac{1}{1 + \epsilon^2} + \frac{1}{\sigma(T)^2} - \frac{1}{1 + \epsilon^2 + \sigma(T)^2}\right) - \frac{1}{2}\log 4\pi e\epsilon^2$$

$$= \frac{1}{2}\log 2(1 + (1 + \epsilon^2)/\sigma^2(T) - (1 + \epsilon^2)/(1 + \epsilon^2 + \sigma^2(T))),$$

which goes to $\infty$ as $\epsilon, \sigma(T) \to 0$. The inequality uses the fact that $h(X+Y) \geq h(X)$ for continuous random variable $X$ and independent variable $Y$.

Now, during the VC inference for source and target speech

$$X^1 = [Z + \Xi^1(1), -1 + \Xi^1(2)], \ X^2 = [Z + \Xi^2(1), 1 + \Xi^2(2)] \tag{44}$$

it can be shown that the maximum likelihood estimator of $X^2$ given $\hat{Z}^2, \hat{G}^2$ is $\hat{X}^{2 \to 2} := [\hat{Z}^2 \hat{G}^2, \hat{G}^2] = [X_T^2(1), \hat{G}^2]$, which is not worse than the diffusion VC learned using Equation 8. However, the converted speech $\hat{X}^{1 \to 2}$ using the same estimator is

$$\hat{X}^{1 \to 2} = [X_T^1(1)\,\mathrm{sign}(\hat{G}^1)\,\mathrm{sign}(\hat{G}^2), \hat{G}^2]. \tag{45}$$

Notice that the truncated conditional means for the first coordinate of the estimator and the true target speech are

$$\mathbb{E}\left[\hat{X}^{1 \to 2}(1)\mathbb{1}_{|\hat{X}^{1 \to 2} - Z| \leq \sigma(T)}\middle| Z\right]$$

$$= (1 - 2Q(1))Z \cdot \mathbb{E}\,\mathrm{sign}(\Xi^1 - 1)\mathbb{E}\,\mathrm{sign}(\Xi^1 + 1) = 2Q(1)(1 - 2Q(1))^2 Z,$$

$$\mathbb{E}\left[X^2(1)\mathbb{1}_{|\hat{X}^{1 \to 2} - Z| \leq \sigma(T)}\middle| Z\right] = (1 - 2Q(1))Z.$$

---

[1]While neither $Z$ nor $G$ are bounded in this example, they can be made so by truncating their tails, though we will not do so to make the calculations simple.

As a result, by the variational characterization of $d_{\text{TV}}$:

$$d_{\text{TV}} = \sup_{|f| \leq 1/2} \mathbb{E}_{q_{X^2}} f(X) - \mathbb{E}_{q_{X^{1 \to 2}}} f(X) \geq$$

$$\mathbb{E}\left[\frac{X^2(1)}{|Z| + \sigma(T)} \mathbb{1}_{|X^2 - Z| \leq \sigma(T)} \,\middle|\, Z\right] - \mathbb{E}\left[\frac{\hat{X}^{1 \to 2}(1)}{2|Z| + 2\sigma(T)} \mathbb{1}_{|\hat{X}^{1 \to 2} - Z| \leq \sigma(T)} \,\middle|\, Z\right] =$$

$$\frac{(1 - 2Q(1))(1 - 2Q(1)(1 - 2Q(1)))Z}{2|Z| + 2\sigma(T)} \xrightarrow{\sigma(T) \to 0} (1/2 - Q(1))(1 - 2Q(1)(1 - 2Q(1))) \approx 0.27,$$

where $Q(\cdot)$ is the tail distribution of a standard Gaussian variable.

Note that in this case, our choice of $\hat{Z}_T$ fails to be disentangled from $\hat{G}$ when the true content variable $Z$ is known, since

$$I(\hat{Z}_T; \hat{G}|Z) = \mathbb{E}_{p(z)}[h(\hat{Z}_T|Z = z) - h(\hat{Z}_T|\hat{G}, Z = z)] \geq h(\hat{Z}_T|Z = z) - h(\hat{Z}_T|\hat{G}, Z = z)$$

$$\geq \mathbb{E}_{p(z)}[h(\hat{Z}_T|\hat{G}, Z = z) + d_{\text{B}}(\mathcal{N}(-z, \epsilon^2 + \sigma(T)^2)||\mathcal{N}(z, \epsilon^2 + \sigma(T)^2)) - h(\hat{Z}_T|\hat{G}, Z = z)] =$$

$$\frac{1}{4(\epsilon^2 + \sigma^2(T))} + \frac{1}{2}\log\frac{\epsilon^2 + \sigma^2(T)}{\epsilon\sigma(T)} \xrightarrow{\epsilon, \sigma(T) \to 0} \infty,$$

where $d_{\text{B}}(p, q) := -\log \int \sqrt{p(x)q(x)}dx$ is the Bhattacharyya distance and the lower bound on the entropy of mixture models in Kolchinsky & Tracey (2017) is used.

## E    PROOF OF THEOREM 2

To further the analysis, we make use of the following lemmas.

**Lemma 2.** *For any differentiable probability density $q$,*

$$\mathbb{E}_q \nabla_x \log q(X) = 0. \tag{46}$$

*Proof.* By definition, $\mathbb{E}_q \nabla_x \log q(X) = \mathbb{E}_q \frac{\nabla_x q(X)}{q(X)} = \nabla_x \int_{\mathcal{X}} q(x)dx = 0.$ $\qquad\square$

**Lemma 3.** *The regularizer Equation 20 is minimized by $(\hat{V}, \theta_G^{\hat{V}})$ if and only if*

$$R(\hat{V}) = R(A_G), \hat{V}s_{\theta_G^{\hat{V}}}(g, t) = A_G\nabla_g \log \alpha_t(g).$$

### E.1    PROOF OF LEMMA 3

Let $\theta := \theta_G$ and $\text{grad}(G_t) := \nabla_g \log \alpha_t(G_t), \text{grad}(Z_t) := \nabla_z \log p_{Z_t}(Z_t)$ for notional ease. Then by definition,

$$L_{\text{reg}}(\theta, V)$$

$$=\mathbb{E}_{t,q_{\alpha,t}}\|Vs_\theta(G_t) - A_G\text{grad}(G_t) - A_Z\text{grad}(Z_t)\|^2$$

$$=\mathbb{E}_{t,q_{\alpha,t}}\|A_Z A_Z^\top Vs_\theta(G_t) - A_Z\text{grad}(Z_t)\|^2 + \mathbb{E}_{t,q_{\alpha,t}}\|A_Z A_Z^\top Vs_\theta(G_t) - A_G\text{grad}(G_t)\|^2$$

$$=\mathbb{E}_{t,q_{\alpha,t}}\|\text{proj}_{R(A_Z)}Vs_\theta(G_t) - A_Z\text{grad}(Z_t)\|^2 + \mathbb{E}_{t,q_{\alpha,t}}\|\text{proj}_{R(A_G)}Vs_\theta(G_t) - A_G\text{grad}(G_t)\|^2.$$

For the first term, use the fact $Z_t \perp\!\!\!\perp G_t$,

$$\mathbb{E}_{t,q_{\alpha,t}}\|\text{proj}_{R(A_Z)}Vs_\theta(G_t) - A_Z\text{grad}(Z_t)\|^2$$

$$\geq\mathbb{E}_{t,q_{\alpha,t}}\|\mathbb{E}\left[A_Z\text{grad}(Z_t)|\,G_t\right] - A_Z\text{grad}(Z_t)\|^2 = \mathbb{E}_{t,q_{\alpha,t}}\|\mathbb{E}_{q_{\alpha,t}}\text{grad}(Z_t) - \text{grad}(Z_t)\|^2$$

$$=\mathbb{E}_{t,q_{\alpha,t}}\|\text{grad}(Z_t)\|^2,$$

by Lemma 2 and with equality iff $A_Z^\top Vs_\theta(g) = 0, \forall g : \alpha_t(g) > 0$.

For the second term, simply notice that it is nonnegative and equal to 0 iff

$$A_G^\top Vs_\theta(G_t) = \text{grad}(G_t), \forall g : \alpha(g) > 0$$

As a result, for any minimizer $(\hat{\theta}, \hat{V})$,

$$\hat{V}s_{\hat{\theta}}(G_t) = A_G A_G^\top \hat{V}s_{\hat{\theta}}(G_t) + A_Z A_Z^\top \hat{V}s_{\theta\hat{V}}(G_t) = A_G \mathrm{grad}(G_t).$$

Further, notice that

$$0 = \mathbb{E}_{t,q_{\alpha,t}}\|\hat{V}s_{\hat{\theta}}(G_t) - A_Z\mathrm{grad}(Z_t)\|^2 \tag{47}$$

$$= \mathbb{E}_{t,q_{\alpha,t}}[\|\hat{V}s_{\hat{\theta}}(G_t) - \mathrm{proj}_{R(\hat{V})}A_G\mathrm{grad}(G_t)\|^2 + \|\mathrm{proj}_{N(\hat{V}^\top)}A_G\mathrm{grad}(G_t)\|_2^2] \tag{48}$$

$$\geq \mathbb{E}_{t,q_{\alpha,t}}\|\mathrm{proj}_{N(\hat{V}^\top)}A_G\mathrm{grad}(G_t)\|_2^2 \geq \lambda_{\min}\|\mathrm{proj}_{N(\hat{V}^\top)}A_G\|_F^2, \tag{49}$$

by Assumption 3. Therefore $\|\mathrm{proj}_{N(\hat{V}^T)}A_G\|_F = 0$, which implies $R(A_G) \subseteq R(\hat{V})$. Further, consider the fact that $\mathrm{rank}(V) \leq \mathrm{rank}(A_G)$, we conclude $R(\hat{V}) = R(A_G)$.

### E.2 MAIN PROOF

First, set $\lambda$ large enough so that $\tilde{L}_{\mathrm{match}} \approx \lambda L_{\mathrm{reg}}$, then by Lemma 3, any minimizer $(\hat{\theta}_G, \hat{V})$ of $L_{\mathrm{reg}}$ satisfies

$$\hat{V}s_{\hat{\theta}_G}(G_t) = A_G \nabla_g \log \alpha_t(G_t).$$

Plug this into $L_{\mathrm{match}}$ yields

$$L_{\mathrm{match}}(\theta_Z, \hat{\theta}_G, U, \mathrm{proj}_{\mathcal{O}}U, \hat{V})$$

$$= \mathbb{E}_{t,q_{\alpha,t}}\|Us_{\theta_Z}(\mathrm{proj}_{\mathcal{O}}U^\top X_t) - A_Z\mathrm{grad}(Z_t)\|_2^2$$

$$= \mathbb{E}_{t,q_{\alpha,t}}[\|Us_{\theta_Z}(\mathrm{proj}_{\mathcal{O}}U^\top X_t) - \mathrm{proj}_{R(U)}A_Z\mathrm{grad}(Z_t)\|^2 + \|\mathrm{proj}_{N(U^\top)}A_Z\mathrm{grad}(Z_t)\|^2]$$

$$\geq \mathrm{Tr}[\mathrm{proj}_{N(U^\top)}A_Z\mathrm{grad}(Z_t)\mathrm{grad}(Z_t)^\top A_Z\mathrm{proj}_{N(U^\top)}]$$

$$\geq \lambda_{\min}\|\mathrm{proj}_{N(U^\top)}A_Z\|_F^2 \geq 0,$$

where the second-to-last inequality uses Assumption 3. Further, notice that the last equality is achieved by $(\hat{U}, \hat{\theta}_Z)$ iff

$$\hat{U}s_{\hat{\theta}_Z}(\mathrm{proj}_{\mathcal{O}}\hat{U}^\top X_t) = \mathrm{proj}_{R(\hat{U})}A_Z\mathrm{grad}(Z_t), \quad \|\mathrm{proj}_{N(\hat{U}^\top)}A_Z\|_F = 0 \implies R(A_Z) \subseteq R(\hat{U}).$$

Combined with the fact $\mathrm{rank}(\hat{U}) \leq \mathrm{rank}(A_Z)$, we have $R(\hat{U}) = R(A_Z)$. Using this fact, we then conclude that

$$\hat{U}s_{\hat{\theta}_Z}(\mathrm{proj}_{\mathcal{O}}\hat{U}^\top X_t) = \mathrm{proj}_{R(\hat{U})}A_Z\mathrm{grad}(Z_t) = A_Z\mathrm{grad}(Z_t).$$

## F PROOF OF THEOREM 3

To begin, notice that the gradient flow equation for the speaker subspace in Equation 22 is simply

$$\dot{V} = -\nabla_V \mathbb{E}_{t,q_{\alpha_t}}\|(V - A_G)\mathrm{grad}(G_t)\|^2 = -2(V - A_G)\mathbb{E}_{t,q_{\alpha_t}}\mathrm{grad}(G_t)\mathrm{grad}(G_t)^\top.$$

Let $E(r) := \|V^{(r)} - A_G\|_F^2$, then

$$\dot{E}(r) = 2\,\mathrm{Tr}[\dot{V}(V - A_G)^\top]$$

$$= -4\,\mathrm{Tr}[(V - A_G)\mathbb{E}_{t,q_{\alpha_t}}\mathrm{grad}(G_t)\mathrm{grad}(G_t)^\top(V - A_G)^\top]$$

$$\leq -4\lambda_{\min}\|V - A_G\|_F^2 = -4\lambda_{\min}E(r),$$

where the inequality uses Assumption 3. As a result, we have

$$E(r) \leq E(0)\exp(-4\lambda_{\min}r) \xrightarrow{r\to\infty} 0 \implies \hat{V} = A_G.$$

Plug $\hat{V}$ into the Equation 23 and let $\hat{Z}_t = \mathrm{sg}(\mathrm{proj}_{\mathcal{O}}(U)^\top X_t)$ and $\mathrm{grad}(\hat{Z}_t) := \nabla_{\hat{z}}\log p_{\hat{Z}_t}(\hat{Z}_t)$, we obtain

$$\dot{U} = -\nabla_U \mathbb{E}_{t,q_{\alpha_t}}\|Us_{\theta_Z^U}(\mathrm{sg}(\mathrm{proj}_{\mathcal{O}}(U)^\top X_t)) - A_Z\mathrm{grad}(Z_t)\|_2^2$$

$$= -2\mathbb{E}_{t,q_{\alpha_t}}(Us_{\theta_Z^U}(\hat{Z}_t) - A_Z\mathrm{grad}(Z_t))s_{\theta_Z^U}(\hat{Z}_t)^\top$$

$$= -2\mathbb{E}_{t,q_{\alpha_t}}(U\mathrm{grad}(\hat{Z}_t) - A_Z\mathrm{grad}(Z_t))\mathrm{grad}(\hat{Z}_t)^\top.$$

Consider the function $F(r) := \mathbb{E}_{t,q_{\alpha,t}} \|U^{(r)} \text{grad}(\hat{Z}_t^{(r)}) - A_Z \text{grad}(Z_t)\|_F^2$, and notice that

$$\dot{F}(r) = 2 \text{Tr}[\text{grad}(\hat{Z}_t)^\top \dot{U}^\top (U \text{grad}(\hat{Z}_t) - A_Z \text{grad}(Z_t))]$$

$$= -4\mathbb{E}_{t,t',q_{\alpha,t},q_{\alpha,t'}} \text{grad}(\hat{Z}_t)^\top \text{grad}(\hat{Z}_{t'}) \text{Tr}[(U \text{grad}(\hat{Z}_t) - A_Z \text{grad}(Z_t))^\top (U \text{grad}(\hat{Z}_{t'}) - A_Z \text{grad}(Z_{t'}))]$$

$$= -4\|U\mathbb{E}_{t,q_{\alpha,t}} \text{grad}(\hat{Z}_t) \text{grad}(\hat{Z}_t)^\top - A_Z \mathbb{E}_{t,q_{\alpha,t}} \text{grad}(\hat{Z}_t) \text{grad}(Z_t)^\top\|_F^2.$$

By Assumption 3,

$$\|U\mathbb{E}_{t,q_{\alpha,t}} \text{grad}(\hat{Z}_t) \text{grad}(\hat{Z}_t)^\top - A_Z \mathbb{E}_{t,q_{\alpha,t}} \text{grad}(\hat{Z}_t) \text{grad}(Z_t)^\top\|_F$$

$$\geq \|U\mathbb{E}_{t,q_{\alpha,t}} \text{grad}(\hat{Z}_t) \text{grad}(\hat{Z}_t)^\top - A_Z A_Z^\top U\mathbb{E}_{t,q_{\alpha,t}} \text{grad}(\hat{Z}_t) \text{grad}(\hat{Z}_t)^\top\|_F$$

$$\geq \lambda_{\min} \|\text{proj}_{N(A_Z^\top)} U\|_F.$$

Therefore, any stationary point of Equation 23 satisfies

$$\|\text{proj}_{N(A_Z)^\top} U\|_F^2 = 0 \implies R(U) \subset R(A_Z) \implies R(\text{proj}_{\mathcal{O}} U) \cap R(U) \neq \emptyset.$$

Therefore, by Assumption 4,

$$\|U\mathbb{E}_{t,q_{\alpha,t}} \text{grad}(\hat{Z}_t) \text{grad}(\hat{Z}_t)^\top - A_Z \mathbb{E}_{t,q_{\alpha,t}} \text{grad}(\hat{Z}_t) \text{grad}(Z_t)^\top\|_F$$

$$\geq \|\text{proj}_{R(U)} A_Z \mathbb{E}_{t,q_{\alpha,t}} \text{grad}(\hat{Z}_t) \text{grad}(Z_t)^\top - A_Z \mathbb{E}_{t,q_{\alpha,t}} \text{grad}(\hat{Z}_t) \text{grad}(Z_t)^\top\|_F$$

$$\geq \rho_{\min} \|\text{proj}_{N(U^\top)} A_Z\|_F,$$

for some $\rho_{\min} > 0$. Consequently, for such any stationary point $\hat{U}$,

$$\dot{F}(r) = 0 \implies \|\text{proj}_{N(\hat{U}^\top)} A_Z\|_2 = 0 \implies R(A_Z) \subseteq R(U).$$

Combining with the fact that $R(U) \subseteq R(A_Z)$ yields $R(\hat{U}) = R(A_Z)$.

# G    PROOF OF THEOREM 4

Since the majority vote classifier makes a mistake only if more than half of the single-speaker probability weights are not on the correct label, then the classifier loss on the test set $(X', Y') \sim P_\beta$ is

$$L_{P_\beta}^{\text{soft}}(f_{\text{mv}}^{\text{soft}}) = \Pr[f_{\text{mv}}^{\text{soft}}(X') \neq Y'] \leq \Pr[\mathbb{E}_{G \sim \rho} f^G(Y'|X') < 1/2] =: \Pr[W_\rho^{\text{soft}}(X', Y') \geq 1/2], \tag{50}$$

where $W_\rho^{\text{soft}}(x, y) := 1 - \mathbb{E}_{G \sim \rho} f^G(y|x)$ is the proportion of probability weights assigned to incorrect labels. We need the concept of *competence* Theisen et al. (2023) to random classifiers in Appendix G. We provide the following generalized definition of competence and the definition of EIR.

**Definition 6.** *For speaker embedding distribution $\rho$ and the test set $(X, Y) \sim P_\beta$, single-speaker classifiers are* competent *if for any $t \in [0, 1/2]$,*

$$\Pr_{P_\beta}[W_\rho^{\text{soft}}(X, Y) \in [t, 1/2)] \geq \Pr_{P_\beta}[W_\rho^{\text{soft}}(X, Y) \in [1/2, 1 - t]]. \tag{51}$$

**Definition 7.** *For data distribution $P$ and single-speaker classifiers $f^G$, $G \sim \rho$ such that $\mathbb{E}_{G \sim \rho} L_P(f^G) \neq 0$, the* ensemble improvement rate *is defined as*

$$\text{EIR}(f_{\text{mv}}) = \frac{\mathbb{E}_{G \sim \rho} L_P(f^G) - L_P(f_{\text{mv}})}{\mathbb{E}_{G \sim \rho} L_P(f^G)}. \tag{52}$$

Using Assumption 6 and replacing the hard votes with soft ones in the proofs of Lemma 2 and part of Theorem 2 in Theisen et al. (2023), we can prove the following lemma.

**Lemma 4.** *If random classifier $Y^G \sim f^G(Y|X)$ is competent for speaker embedding distribution $\rho$ and the test distribution $P_\beta$,*

$$\Pr_{P_\beta}[W_\rho^{\text{soft}}(X, Y) \geq 1/2] \leq 2\mathbb{E}_{P_\beta}[W_\rho^{\text{soft}}(X, Y)^2]. \tag{53}$$

We proceed to bound the second moment of $W_\rho^{\mathrm{soft}}$ via the following lemma, analogous to Lemma 4 in Theisen et al. (2023).

**Lemma 5.** *If random classifier $Y^G \sim f^G(Y|X)$ is competent for speaker embedding distribution $\rho$ and test data distribution $P_\beta$,*

$$\mathbb{E}_{P_\beta}[W_\rho^{\mathrm{soft}}(X,Y)^2] \leq \frac{2(|\mathcal{Y}|-1)}{|\mathcal{Y}|}\left(\mathbb{E}_{G\sim\rho}L_{P_\beta}^{\mathrm{soft}}(f^G) - \frac{1}{2}\mathbb{E}_{G\sim\rho,G'\sim\rho}D^{\mathrm{soft}}(f^G,f^{G'})\right),$$

*where $D^{\mathrm{soft}}(h,h') := \mathbb{E}_{X\sim P}\mathbb{E}_{\hat{Y}\sim h(\cdot|X),\hat{Y}'\sim h(\cdot|X)}\mathbb{1}[\hat{Y}\neq\hat{Y}']$ is the* disagreement rate *between random classifiers $\hat{Y}$ and $\hat{Y}'$.*

*Proof.* Let $(X,Y) \sim P_\beta$ be a ground truth feature-label pair from the test set and $h_Y(X) := \mathbb{E}_{G\sim\rho}[f^G(Y|X)]$. Moreover, let $q_Y^G(\cdot|X)$ be the true posterior probability of $Y$ given the original speech $X$. Then for $G = g$, the conditional error rate of the random classifier $Y^g$ is

$$\Pr_{P_\beta}[Y^g \neq Y|X] = 1 - \mathbb{E}_{q_Y^g(Y|X)}f^G(Y|X),$$

and the conditional disagreement rate of two independent random classifiers $Y^g$ and $Y^{g'}$ is

$$\Pr_{P_\beta}[Y^g \neq Y^{g'}|X] = 1 - \sum_j h^g(j|X)f^{G'}(j|X) = \sum_j f^G(j|X)(1 - f^{G'}(j|X)).$$

Then by definition of $W_\rho$,

$$W_\rho^2(X,Y) = 1 - h_Y(X) - h_Y(X)(1 - h_Y(X))$$
$$= \underbrace{1 - h_Y(X)}_{(1)} - \underbrace{\sum_j h_j(X)(1-h_j(X))}_{(2)} + \underbrace{\sum_{k\neq Y} h_k(X^G)(1-h_k(X^G))}_{(3)}.$$

Taking the expectation over $X,Y$ for the first term yields

$$1 - \mathbb{E}_{P_\beta}h_Y(X) = 1 - \mathbb{E}_{G\sim\rho}\mathbb{E}_{P_\beta^G}f^G(Y|X)$$
$$= 1 - \mathbb{E}_{G\sim\rho}\Pr_{P_\beta^G}[Y^G = Y] = \Pr_{P_\beta}[Y^G \neq Y].$$

To lower bound term (2), simply notice that

$$\sum_j h_j(X)(1-h_j(X)) = \mathbb{E}_{G\sim\rho,G'\sim\rho}f^G(j|X)(1-f^{G'}(j|X)) = \mathbb{E}_{G\sim\rho,G'\sim\rho}[D(f^G,f^{G'})|X].$$

Finally, to bound term (3), we can apply a similar bound in Theisen et al. (2023) by maximizing over $h_k(X^G)$ to conclude that

$$(3) \leq \frac{|\mathcal{Y}|-2}{|\mathcal{Y}|-1}(1-h_Y(X)) + \frac{1}{K-1}h_Y(X)(1-h_Y(X)).$$

As a result, we have

$$(3) - (2) \leq \frac{|\mathcal{Y}|-2}{|\mathcal{Y}|-1}\times(1) + \frac{1}{|\mathcal{Y}|-1}\times((2)-(3)) - \mathbb{E}_{G\sim\rho,G'\sim\rho}D(f^G,f^{G'})$$

$$\implies W_\rho^2(X,Y) = (1) - (2) + (3) \leq \frac{2(|\mathcal{Y}|-1)}{|\mathcal{Y}|}\times\left((1) - \frac{1}{2}\mathbb{E}_{G\sim\rho,G'\sim\rho}[D(f^G,f^{G'})|X]\right).$$

Marginalizing over $X,Y$ yields the result. $\square$

Note that constraining $f^G(Y|X)$ to be deterministic recovers Lemma 4 of Theisen et al. (2023).

Now, consider the case for the hard majority vote classifier where the single-speaker classifiers are deterministic and competent. To this end, applying Lemma 4 and 5 with $f^G(y|x) = \mathbb{1}[f^G(x) = y]$ yields

$$L_{P_\beta}(f_{\mathrm{mv}}) \leq \frac{4(|\mathcal{Y}|-1)}{|\mathcal{Y}|}\left(\mathbb{E}_{G\sim\rho}L_{P_\beta}(f^G) - \frac{1}{2}\mathbb{E}_{G\sim\rho,G'\sim\rho}\Pr_{P_\beta}[f^G(X)\neq f^{G'}(X)]\right). \quad (54)$$

To lower bound $\Pr[f^G(X) \neq f^{G'}(X)]$, notice

$$\Pr_{P_\beta}[f^G(X) \neq f^{G'}(X)]$$

$$= \Pr_{P_\beta}\left[\arg\max_y f^G(y|X) \neq \arg\max_{y'} f^{G'}(y'|X)\right]$$

$$= 1 - \Pr_{P_\beta}\left[\arg\max_j f^G(j|X) = \arg\max_{j'} f^G(j'|X)\right].$$

Further, by Assumption 5, and suppose without loss of generality, $\arg\max_j f^G(j|X) = \arg\max_{j'} f^G(j'|X) = 1$,

$$\left|\log \frac{f^{G'}(1|X)}{f^G(1|X)}\right| \leq \log \frac{1 - \delta_2}{\frac{1}{|\mathcal{Y}|} + \delta_1},$$

$$\forall j > 1, \left|\log \frac{f^{G'}(j|X)}{f^G(j|X)}\right| \leq \log \frac{1 - \sum_{k \neq j} f^{G'}(k|X)}{\delta_3} \leq \log \frac{\frac{|\mathcal{Y}|-1}{\mathcal{Y}} - \delta_1 - (\mathcal{Y}-2)\delta_3}{\delta_3}.$$

As a result,

$$D_{\mathrm{KL}}(f^{G'}(1|X)||f^G(1|X))$$

$$\leq f^{G'}(1|X) \log \frac{1 - \delta_2}{\frac{1}{|\mathcal{Y}|} + \delta_1} + (1 - f^{G'}(1|X)) \log \frac{\frac{|\mathcal{Y}|-1}{\mathcal{Y}} - \delta_1 - (\mathcal{Y}-2)\delta_3}{\delta_3}$$

$$\leq (1 - \delta_2) \log \frac{1 - \delta_2}{\frac{1}{|\mathcal{Y}|} + \delta_1} + \left(\frac{|\mathcal{Y}| - 1}{|\mathcal{Y}|} - \delta_1\right) \log \frac{\frac{|\mathcal{Y}|-1}{\mathcal{Y}} - \delta_1 - (\mathcal{Y}-2)\delta_3}{\delta_3} = \delta^*.$$

Moreover, since random classifiers $f^G(Y|X)$ are $(\kappa_1, \kappa_2)$-speaker distorted,

$$\Pr_{P_\beta}\left[\arg\max_j f^G(j|X) = \arg\max_{j'} f^G(j'|X)\right]$$

$$\leq \Pr_{P_\beta}[D_{\mathrm{KL}}(f^{G'}(1|X)||f^G(1|X)) \leq \delta^*] = \Pr_{P_\beta}[\kappa_1 \|G' - G\|_2 \leq \delta^*]$$

$$= \Pr_{P_\beta}\left[\|G' - G\|_2 \leq \left(\frac{\delta^*}{\kappa_1}\right)^{1/\kappa_2}\right].$$

Let $H := \|G - G'\|_2$, since $G, G'$ are independent, identically-distributed, isotropic sub-gaussian random vectors by Assumption 2, $H - \mathbb{E}H$ is a sub-gaussian random variable with sub-gaussian norm

$$\|H - \mathbb{E}H\|_{\phi_2} \leq C,$$

for some constant $C > 0$ independent of the dimension of $G$ according to Theorem 6.3.2 in Vershynin (2018). Further, by the properties of sub-gaussian random variables,

$$\Pr_{P_\beta}\left[\|G' - G\|_2 \leq \left(\frac{\delta^*}{\kappa_1}\right)^{1/\kappa_2}\right] \leq \Pr_{P_\beta}\left[|H - \mathbb{E}H| \geq \left|\mathbb{E}H - \left(\frac{\delta^*}{\kappa_1}\right)^{1/\kappa_2}\right|\right]$$

$$\leq 2\exp\left(-C'\left|\mathbb{E}H - \left(\frac{\delta^*}{\kappa_1}\right)^{1/\kappa_2}\right|^2\right) = 2\exp(-C'|c\sqrt{d_G} - (\delta^*/\kappa_1)^{1/\kappa_2}|^2),$$

for some dimension-independent constants $C', c > 0$. $\Pr[f^G(X) \neq f^{G'}(X)]$ can be then lower bounded by

$$1 - 2\exp(-C'|c\sqrt{d_G} - (\delta^*/\kappa_1)^{1/\kappa_2}|^2),$$

which can be plugged into equation 54 to obtain

$$\text{EIR}^{\text{hard}} = \frac{L_{P_\beta}(h_{\text{mv}}^{\text{hard}}) - \mathbb{E}_\rho L_{P_\beta}(f^G)}{\mathbb{E}_\rho L_{P_\beta}(f^G)}$$

$$\leq \frac{3|\mathcal{Y}| - 4}{|\mathcal{Y}|} - \frac{2|\mathcal{Y}| - 2}{|\mathcal{Y}|} \frac{1 - 2\exp(-C'|c\sqrt{d_G} - (\delta^*/\kappa_1)^{1/\kappa_2}|^2)}{\mathbb{E}_\rho L_{P_\beta}(f^G)}.$$

For the soft majority vote classifier, we proceed again to bound the terms in Lemma 5 separately. Fix any $G$ and by Assumption 5,

$$L_{P_\beta}^{\text{soft}}(f^G) = 1 - \mathbb{E}_{P_\beta} f^G(Y|X)$$

$$= 1 - \mathbb{E}_{P_\beta} f^G(Y|X)\mathbb{1}[f^G(X) = Y] - \mathbb{E}_{P_\beta} f^G(Y|X)\mathbb{1}[f^G(X) \neq Y]$$

$$\leq 1 - (1/|\mathcal{Y}| + \delta_1)(1 - L_{P_\beta}(f^G)) - \delta_3 L_\beta(f^G) = \left(\frac{|\mathcal{Y}| - 1}{|\mathcal{Y}|} - \delta_1\right) + \left(\frac{1}{|\mathcal{Y}|} + \delta_1 - \delta_3\right) L_\beta(f^G)$$

$$= a_1 L_\beta(f^G) + \frac{|\mathcal{Y}| - 1}{|\mathcal{Y}|} - \delta_1.$$

Similarly, for the disagreement rate,

$$D^{\text{soft}}(f^G, f^{G'}) = 1 - \mathbb{E}_{P_\beta} \sum_j f^G(j|X) f^{G'}(j|X)$$

$$= 1 - \mathbb{E}_{P_\beta} \sum_j f^G(j|X) f^{G'}(j|X)\mathbb{1}[f^G(X) \neq f^{G'}(X)] - \mathbb{E}_{P_\beta} \sum_j f^G(j|X) f^{G'}(j|X)\mathbb{1}[f^G(X) = f^{G'}(X)]$$

$$\geq 1 - 2(1/|\mathcal{Y}| + \delta_1)\delta_3 \Pr[f^G(X) \neq f^{G'}(X)] - ((1 - \delta_2)^2 + \delta_2^2)\Pr[f^G(X) = f^{G'}(X)]$$

$$= 1 - 2(1/|\mathcal{Y}| + \delta_1)\delta_3 - ((1 - \delta_2)^2 + \delta_2^2 - 2(1/|\mathcal{Y}| + \delta_1)\delta_3)\Pr[f^G(X) = f^{G'}(X)]$$

$$= 1 - 2(1/|\mathcal{Y}| + \delta_1)\delta_3 - (2(1 - \delta_2)^2 + 2\delta_2^2 - 4(1/|\mathcal{Y}| + \delta_1)\delta_3)\exp(-C'|c\sqrt{d_G} - (\delta^*/\kappa_1)^{1/\kappa_2}|^2)$$

$$= 1 - 2(1/|\mathcal{Y}| + \delta_1)\delta_3 - a_3 \exp(-C'|c\sqrt{d_G} - (\delta^*/\kappa_1)^{1/\kappa_2}|^2),$$

where the last equality use the upper bound on $\Pr[f^G(X) = f^{G'}(X)]$ derived earlier. Plug the bounds on $L_{P_\beta}^{\text{soft}}(f^G)$ and $D^{\text{soft}}(f^G, f^{G'})$ into Lemma 5 and follow similar steps for $\text{EIR}^{\text{hard}}$, we obtain the bound on $\text{EIR}^{\text{soft}}$.

## H    EXPERIMENT DETAILS

### H.1    SYNTHETIC EXPERIMENT

For visualization purposes, we choose the content and speaker subspace to be $d_Z = d_G = 1$ and both the content GMM to be $\frac{1}{2}\mathcal{N}(-0.5, 0.01) + \frac{1}{2}\mathcal{N}(0.5, 0.01)$ and the speaker GMM to be $\frac{1}{2}\mathcal{N}(-1, 0.01) + \frac{1}{2}\mathcal{N}(1, 0.01)$. We choose a small number of mixtures to avoid bad local optima during training. We adopt a similar for disentanglement experiments but increase the dimension to 5. More details are included in Appendix H. For the disentanglement experiment, we choose the subspace dimension to be $d_Z = d_G = 5$ and sample the content variable via $Z \sim \frac{1}{2}\mathcal{N}(\mu_1^Z, 0.01 \cdot \mathbf{I}_{d_Z}) + \frac{1}{2}\mathcal{N}(\mu_2^Z, \sigma_0^2 \mathbf{I}_{d_Z})$ and the speaker variable via $G \sim \frac{1}{2}\mathcal{N}(\mu_1^G, \sigma_0^2 \cdot \mathbf{I}_{d_G}) + \frac{1}{2}\mathcal{N}(\mu_2^G, \sigma_0^2 \cdot \mathbf{I}_{d_G})$, where we randomly sample the mixture centers via $\mu_k^Z \sim \text{Unif}[-0.5, 0.5]^{d_Z}$ and $\mu_k^G \sim \text{Unif}[-2.5, 2.5]^{d_G}$ and set $\sigma_0 = 0.01$. We generate 2000 samples for both datasets.

We then train a unconditional and a conditional score networks to learn to perform disentanglement and zero-shot voice conversion on the synthetic data. For the unconditional score network, we use a

simple neural net with residual connection:

$$\hat{z}(x) := U^\top x, \tag{55}$$

$$\hat{s}^Z_{\mu,U}(z,t) := \frac{1}{\alpha^2(t)\sigma_0^2 + \sigma^2(t)}[\tanh(\alpha(t)\mu^\top z/(\alpha^2(t)\sigma_0^2 + \sigma^2(t)))\mu - z], \tag{56}$$

$$\hat{s}^G_{\nu,V}(g,t) := \frac{1}{\alpha^2(t)\sigma_0^2 + \sigma^2(t)}[\tanh(\alpha(t)\nu^\top g/(\sigma_0^2 + \sigma^2(t)))\nu - g], \tag{57}$$

$$\hat{s}_{\mu,\nu,U,V}(\hat{z}(x),g,t) := \frac{1}{\sigma^2(t)}\left[U\hat{s}^Z_{\mu,U}(\hat{z}(x),t) + V\hat{s}^G_{\nu,V}(g,t)\right], \tag{58}$$

where $U^\top U = I_{d_Z}$ and $V^\top V = I_{d_G}$ are matrices with orthogonal columns. The neural net is parameterized to match the score function of GMM Shah et al. (2023) in both subspaces. We train the models for 10,000 steps with an Adam Kingma & Ba (2015) optimizer with learning rate $10^{-3}$ and batch size equal to the entire training set. To enforce the orthogonality constraint, we define a customized layer using the `geoopt` package for Riemannian optimization on the Stiefel manifold. To ensure convergence, we pretrained the speaker score network $\hat{s}^G$. We experiment with various noising schedules, including the variance exploding (VE), vanilla variance preserving (VP) Ho et al. (2020), sub-VP Song et al. (2021) and cosine VP Nichol & Dhariwal (2021). The detailed schedule hyperparameters are listed in Table 3 and are chosen

| Name | $\alpha(t)$ | $\sigma^2(t)$ |
|---|---|---|
| VE | 1 | $\frac{25^{2t}-1}{2\log 25}$ |
| VP | $e^{-0.05t-4.975t^2}$ | $1 - e^{-0.1t-9.95t^2}$ |
| sub-VP | $e^{-0.05t-4.975t^2}$ | $(1 - e^{-0.1t-9.95t^2})^2$ |
| VP (cosine) | $e^{-\frac{t}{2}-\frac{1}{\pi}\sin(\frac{t}{2})}$ | $1 - e^{-t-\frac{2}{\pi}\sin(\frac{t}{2})}$ |

Table 3: The default noise schedule hyperparameters for the synthetic data experiments. Continuous time ($t \in [10^{-5}, 1]$) is used in the expression.

based on rules of thumbs in Song & Ermon (2020); Song et al. (2021). Once the unconditioned score network is trained, we then used the learned subspace $\hat{U}$ to create an auxiliary variable $a := \hat{z}(x_0) = \hat{U}^\top x_0$, where $x_0$ is the clean speech feature at time 0 and train another *conditional* score network with $A$ as an additional input as:

$$\hat{s}_{\mu,\nu,U,V}(\hat{z}(x_t),g,t|a) = \frac{1}{\sigma^2(t)}\left[U\frac{\alpha(t)a - \hat{z}(x_t)}{\alpha^2(t)\sigma_0^2 + \sigma^2(t)} + V\hat{s}^G_{\nu,V}(g,t)\right]. \tag{59}$$

For voice conversion experiments, we run the predictor-corrector sampling scheme alternating between Euler-Maruyama method and Langevin MCMC Song et al. (2021) for 500 steps and a signal-to-noise ratio (SNR) parameter of $0.16$.

## H.2 REALISTIC DATA EXPERIMENT

For the IEMOCAP dataset, we use a system available on SpeechBrain Ravanelli et al. (2024) that finetunes on the wav2vec 2.0 backbone Baevski et al. (2020) with a multi-layer perceptron classifier (MLP) Wang et al. (2021b). The classifier is trained using Adam optimizer for 30 epochs with a batch size of 4 and a learning rate of $10^{-4}$ for the MLP and the $10^{-5}$ learning rate for wav2vec 2.0 weights. The system is then evaluated using the standard classification accuracy metric and 5-fold cross validation Busso et al. (2008); Ma et al. (2024). For each fold, we use all 8 speakers from the training set as target speakers.

On the ALS and ADReSS, we use whisper-medium Radford et al. (2023) features, as they have shown to be the most effective for speech impairment classification Wang et al. (2024). To avoid unfair comparison, We concatenate hidden representations over *all* layers of the whisper-medium encoder rather than selecting a particular layer and perform mean pooling over the frame-level features. For both datasets, we follow the standard splits used in previous works Vieira et al. (2022) to have no overlaps between speaker in the training and test sets. And for both datasets, we use the 15 most frequent speakers from the training set as target speakers for the VC to achieve maximize conversion quality via better speaker representation.

We apply the VCs in mostly a zero-shot, plug-and-play fashion, and leave finetuning to specific datasets for future works. For the Diff-VC, we use the publicly available score network and vocoder checkpoints trained on LibriTTS and adopt the original inference hyperparameter settings for all experiments. Similarly, we use the pretrained models and for other VC models. Further, we use a

maximum of 120 second speech from the target speaker to compute the target speaker embedding for all models except KNN-VC, where we use all the target speech as the pool for nearest neighbor search. We also compare VC adaptation with common data augmentation technique such as pitch shifting, where we shift the pitch of all the speech utterances to equally spaced pitch levels over the $F0$ range of the training speech data with levels equal to the number of target speakers and train separate classifiers for each level as in the case of using VC adaptation.

Table 4 5 6 show the complete results for the realistic dataset experiments.

Table 4: Emotion recognition results on IEMOCAP. 8 speakers in the training set of each fold are used as target speakers.

| VC type | Voting type | Accuracy | | | | | |
|---|---|---|---|---|---|---|---|
| | | 1 | 2 | 3 | 4 | 5 | Avg. |
| No VC | - | 72.6 | 76.6 | 68.9 | 68.9 | 70.3 | 71.5 |
| Pitch shifting | single (best) | 64.0 | 65.3 | 57.4 | 57.7 | 58.6 | 60.6 |
| | single (avg) | 61.3 | 62.1 | 50.2 | 48.9 | 53.0 | 55.1 |
| | majority | 61.2 | 65.3 | 58.5 | 57.8 | 62.5 | 61.1 |
| | soft majority | 60.8 | 65.4 | 57.8 | 57.5 | 61.5 | 61.1 |
| KNN-VC | single (best) | 71.2 | 75.4 | 68.3 | 71.9 | 69.1 | 70.4 |
| | single (avg) | 69.6 | 72.6 | 67.0 | 69.9 | 67.4 | 69.3 |
| | majority | 70.3 | 75.6 | 68.5 | 72.8 | 70.0 | 71.4 |
| | soft majority | 70.3 | 76.1 | 68.5 | 72.8 | 69.9 | 71.5 |
| TriAAN-VC | single (best) | 65.5 | 66.9 | 63.0 | 67.5 | 62.6 | 65.1 |
| | single (avg) | 64.6 | 66.3 | 61.1 | 65.6 | 63.1 | 64.1 |
| | majority | 66.9 | 69.0 | 63.9 | 67.9 | 66.5 | 66.8 |
| | soft majority | 66.6 | 69.9 | 63.6 | 68.8 | 67.0 | 67.2 |
| Diff-VC | single (avg) | 87.0 | 88.3 | 86.2 | 87.6 | 86.1 | 87.0 |
| | single (best) | 94.4 | 94.8 | 94.2 | 95.2 | 92.9 | 94.3 |
| | majority | 97.5 | 96.7 | 95.2 | 98.1 | 94.9 | 96.5 |
| | soft majority | 97.7 | 97.6 | 96.3 | 98.7 | 95.6 | 97.2 |

Table 5: Alzheimer detection results on ADReSS

| VC type | Voting type | Precision | Recall | F1 | Accuracy |
|---|---|---|---|---|---|
| No VC | - | 71.4 | 70.8 | 70.6 | 70.8 |
| Pitch shifting | single (avg) | 71.8 | 71.4 | 71.4 | 71.2 |
| | single (best) | 77.1 | 77.1 | 77.1 | 77.1 |
| | majority | 77.1 | 77.1 | 77.1 | 77.1 |
| | soft majority | 68.8 | 68.8 | 68.7 | 68.8 |
| KNN-VC | single (avg) | 71.8 | 71.5 | 71.4 | 71.5 |
| | single (best) | 80.0 | 79.2 | 79.1 | 79.2 |
| | majority | 79.4 | 79.2 | 79.1 | 79.2 |
| | soft majority | 83.6 | 83.3 | 83.3 | 83.3 |
| TriAAN-VC | single (avg) | 72.5 | 72.4 | 72.4 | 72.4 |
| | single (best) | 75.2 | 75.0 | 75.0 | 75.0 |
| | majority | 77.5 | 77.1 | 77.0 | 77.1 |
| | soft majority | 83.3 | 83.3 | 83.3 | 83.3 |
| Diff-VC | single (avg) | 65.7 | 65.4 | 65.4 | 65.6 |
| | single (best) | 69.4 | 69.4 | 69.4 | 69.4 |
| | majority | 66.7 | 66.7 | 66.7 | 66.7 |
| | soft majority | 72.2 | 70.8 | 70.4 | 70.8 |

Table 6: ALS severity classification results on ALS-TDI with a whisper-medium+SVM classifier

| VC type | Voting type | Precision↑ | Recall↑ | F1↑ |
|---|---|---|---|---|
| No VC | - | 59.8 | 53.7 | 54.9 |
| Pitch shifting | single (avg.) | 60.5 | 54.1 | 55.8 |
| | single (best) | 67.4 | 57.7 | 60.3 |
| | majority | 73.0 | 54.9 | 57.6 |
| | soft majority | 68.4 | 59.0 | 61.5 |
| KNN-VC | single (avg.) | 58.5 | 54.6 | 55.8 |
| | single (best) | 65.7 | 59.5 | 61.7 |
| | majority | 67.9 | 62.9 | 64.8 |
| | soft majority | 51.1 | 49.6 | 49.9 |
| TriAAN-VC | single (avg.) | 60.2 | 54.5 | 55.7 |
| | single (best) | 68.0 | 58.2 | 60.7 |
| | majority | 69.9 | 59.1 | 61.7 |
| | soft majority | 54.1 | 52.8 | 53.3 |
| Diff-VC | single (avg.) | 48.2 | 47.0 | 47.0 |
| | single (best) | 53.0 | 51.0 | 51.2 |
| | majority | 49.8 | 50.9 | 50.3 |
| | soft majority | 50.1 | 48.8 | 49.2 |

