# OpenReview forum: "Can Diffusion Models Disentangle? A Theoretical Perspective"
_ICLR.cc/2025/Conference — ICLR 2025 Conference Withdrawn Submission_

### Official Review · Reviewer_cVqv · 2024-10-27

**Soundness:** 2
**Presentation:** 1
**Contribution:** 2
**Rating:** 3
**Confidence:** 4

**Summary:**

This paper studies theoretical aspects of disentanglement in diffusion models. It provides the theory strongly relying on the notion of *implicit bottleneck variable* the authors introduce and assumptions under which voice conversion (chosen as a typical disentanglement problem) can be performed with a good quality as measured by the notion *ϵ-semantically matched VC* also introduced by the authors. Next, a special case of linear subspace models is studied and a novel regularization term is introduced that hepls a diffusion model learn disentangled representation. The theoretical section ends with the analysis of the ways classification with train-test domain mismatch can be improved by means of *adapt-and-vote* technique where adaptation stage is performed by an imperfect voice conversion model. The experimental section contains experiments on synthetica data with linear subspace diffusion models and three experiments on real voice data illustrating how classification tasks can benefit from domain adaptation with 4 voice conversion techniques including diffusion-based one.

**Strengths:**

The main topic of this paper is the problem of disentanglement in diffusion models and possible ways to enforce it. This is a very important topic for generative modeling.

**Weaknesses:**

First of all, I think that the presentation style of the theoretical part of this paper leaves much to be desired: (i) very often notation is used before it is introduced; moreover, in many cases this notation is used without being introduced/defined anywhere in the main part of the paper; (ii) sometimes the notation is inconsistent; (iii) lot's of typos, and I do not mean just textual ones, there are typos in the formulae in the main part of the paper. These issues make it really hard to understand and evaluate theoretical contribution of the paper. Below I list some examples of such issues occurring only in Section 2 and subsection 2.1 (it's just two and a half pages):
1. In lines 068-069 $Y$ stands for classification tasks, in lines 070-071 $Y$ is referred to as labels corresponding to classification tasks.
2. Content random variables $Z$ come from distribution with density $q_{Z}$ in many places in the paper (e.g. line 068), but in line 182 (Assumption 2, item 5) the content density is denoted by $p_{Z}$.
3. Speech features $X\sim q_{\gamma}$ in line 064 and speaker identity $G\sim\gamma$ in line 069. Neither $\gamma$ nor $q_{\gamma}$ are defined. If $\gamma$ is distribution of the random variable $G$ corresponding to speaker identity, then $q_{\gamma}$ should be its density. Why then do speech features $X$ come from the distribution with the density $q_{\gamma}$?
4. Line 077 (Assumption 1, item 3) $d$ is not defined. Functions $I$ and $h$ are not defined either. I understand that they denote mutual information and entropy, but it's better to state that explicitly, especially if you also use the same notation $h$ in line 267 for a multi-class classifier.
5. It's better to be precise and call $B_{t}$ a Wiener process rather than just some Markov process (line 086-087), because later you explicitly use that diffusion process (2) is the one relying on Gaussian noise when you write down Formula (5).
6. In Formula (2) there should be just $X_{t}$ instead of $X_{t}^{\leftarrow}$.
7. Lines 089-090: word "wehe".
8. Formula (3): there should have been $f(X_{t}^{\leftarrow}, A, T-t)$ instead of $f(X_{t}^{\leftarrow}, A, t)$ in the drift coeficient of the reverse diffusion similarly to diffusion coefficient $\nu(T-t)$. The same mistake in the drift coefficient can be found in Formula (7) and the one in lines 147-150, while Formula (10) has the right drift term $f(X_{t}^{2\leftarrow 1}, A, T-t)$.
9. In the same Formula (3) there must be $dB_{t}^{\leftarrow}$ instead of $dB_{t}$ since we switch to the reverse-time dynamics.
10. In lines 097-098 $\chi$ is not defined.
11. In lines 121 we start considering speech features $X^{1}\sim q_{\beta,0}$ while before we had speech features with densities $q_{\gamma}$ and $q_{\alpha}$. None of $\alpha$, $\beta$ or $\gamma$ are defined in the paper up to this moment. Am I correct that $\gamma$ are all possible speech features, and $\alpha$ and $\beta$ correspond to speech features for two particular speakers? Moreover, $\gamma$ is used later in the definition of the implicit bottleneck variable (Formula (11)), and it certainly has another meaning than when it is used in the expressions like $q_{\gamma}$.
12. In Formula (9) the drift term should be $f(X_{t}^{1},A,t)dt$ rather than just $-f(X_{t}^{1},A,t)$ without time differential.
13. In Formula (10) $\hat{\theta}$ is not defined.
14. In Formula (10) you use $B_{t}^{\leftarrow}$ which must be Wiener process in reverse-time, but it is never mentioned in the paper.
15. The acronym AEVC in lines 127-128 is not defined. Does it mean "Autoencoder Voice Conversion"?
16. In Formula (11) in the definition of the implicit bottleneck variable (which is key to the whole theoretical framework you've designed) you use undefined notation $G_{<t}$. This indeed makes it very difficult to understand what is going on after in this section.
17. In the formula in lines 147-150 in the third term you should have $d\tau$ instead of $dt$.
18. In Formula (13) an undefined function $\hat{g}$ is used.
19. In Definition 3 you use $d_{TV}$ distance without mentioning what it is.
20. In Assumption 2, item 1 you define $t^*$, but do not use it anywhere in this Assumption.

There is also a mistake in Formula (5): it suggests that the score function is $-(X_{t}-X_{0})/\sigma^{2}(t)$ for all $t$, but this is not true. E.g. in case of Variance Preserving diffusions [1] the score function is $-(X_{t}-a(t)X_{0})/\sigma^{2}(t)$ where $X_{t}|X_{0}\sim a(t)X_{0}+\sigma(t)\mathcal{N}(0,1)$ and $a(t)$ varies in $t$.

The next weakness regards the notion of implicit bottleneck variable. First of all, I feel that the definition itself is ambiguous in the sense that the functions $\eta$, $\zeta$ and $\gamma$, if they exist, are not unique, and they are not even unique up to some linear transform that could preserve "information" contained in, for example, $\zeta(X_{t}^{\leftarrow})$. A simple example is as follows: suppose $X_{t}^{\leftarrow}=|X_{0}^{\leftarrow}|$, then we can have either $\zeta(X_{t}^{\leftarrow})=X_{0}^{\leftarrow}$ and $\eta(z,g)=|z|$, or $\zeta(X_{t}^{\leftarrow})=|X_{0}^{\leftarrow}|$ and $\eta(z,g)=z$. And further assumptions (Assumption 2) rely heavily on properties of $\zeta^{\star}=\zeta(\hat{X}_{<t^{*}}^{\leftarrow})$, e.g. on values like $I(\zeta^{\star};X)$ that are different in the two cases.

The former issue leads to the next concern: we really do not know much about this implicit bottleneck variable $\zeta^{\star}$. What are the sufficient conditions for it to exist? Are items 1 and 4 in the Assumption 2 feasible? This issue is very important, because you build your theory in Section 2 based on the random variable $\zeta^{\star}$ which is not even guaranteed to exist (it corresponds to the case $t^{*}=0$ if I'm not mistaken). Apart from this, I also have some less important considerations which I'll list below as questions.

As for experimental part, I also feel it has serious drawbacks. As far as I understand, the whole theoretical framework derived in Section 2 is tested only on synthetic datasets while the authors claim that they describe the mechanism behind disentanglement in general diffusion models. Moreover, after this claim they start describing their setup in "intuitive" voice conversion terms which may be a bit confusing. If they do so, it would be good to demonstrate some of their findings on at least one example not related to voice conversion.

In the experiments on synthetic datasets, the authors consider linear subspace diffusion models which decompose speech features following Equation (18). The papers you refer to before this equation to justify this decomposition deal with discriminative tasks, but you consider voice conversion, a generative task. I don't know papers that successfully use this type of decomposition in generative tasks, and it seems very unlikely that such a simplified point of view on speech features decomposition can lead to a good quality of generated speech. So, I do not think that the results of these experiments on synthetic datasets can be extrapolated to real voice conversion.

The experiments on realistic datasets correspond to theoretical findings in Section 3 which are actually more related to ensembling theory than to disentanglement in diffusion models. Altough I do not have much experience in ensembling methods, I think that the first conclusion the authors make ("Adding target speakers reduces speaker distortion") is not surprising and can be explained not only by Theorem 4, but by the reduced variance as well: when the number of target speakers for voice conversion increases, the variance introduced by different performance of classifiers on different (imperfectly generated) voices gets lower resulting in improved performance in classification task when some kind of aggregation mechanism is used. Two other conclusions ("Different VC excels at different tasks" and "Tradeoff between classifier accuracy and diversity") are quite interesting, but they can not be explained by the theoretical framework the authors have developed.

Overall, I think that the theory trying to explain disentanglement in diffusion models presented in this paper lacks convincing experimental material, has some issues regarding feasibility of the assumptions crucial for this theory, and the text itself has to be made more readable by getting rid of numerous typos/mistakes/inconsistencies in notation.

[1] Score-Based Generative Modeling through Stochastic Differential Equations, Song et al.

**Questions:**

1. What is the conceptual difference between $A$ and $Z_{t}$ in Formula (4)? They both seem to correspond only to content information. When you define $A$, you refer to [2], but in that paper the score function depended only on $X_{t}$, $A$, $t$ and speaker information, there was not additional bottleneck $Z_{t}=z_{\phi, t}(X_{t})$ instead of $X_{t}$. Also, does this formula mean that $z_{\phi}$ is a separate neural network? Is it supposed to be trained just jointly with the score-matching network $s_{\theta}$?
2. In lines 128 - 130, why do you call $X_{T}$ a time-dependent variable in contrast with time-independent $A$? $X_{T}$ is also independent of diffusion time $t$ and depends only on a fixed time horizon $T$.
3. I do not understand Formula (13). If $a=1$ and $b=2$, then $\hat{X}^{a\to b}$ should be the result of forward and reverse diffusions defined in Equations (9-10), but I cannot see how the Definition 1 of the implicit bottleneck and (9-10) imply Formula (13). What also bothers me is that you define the implicit bottleneck variable by considering the critical timestep $t^{*}$ for the reverse process $X_{t}^{\leftarrow}$, and it is the diffusion time for reverse-time processes. But in Formula (13) you use expressions like $\zeta(X_{<t^{\star}}^{a})$, but here time $t$ bounded by the critical $t^{\star}$ already corresponds to forward-time process. Is it correct? To increase presentation clarity, I would recommend to use different notation for forward and reverse timesteps (e.g. $t$ and $\tau$).
4. Regarding Formula (13), I also don't understand what is $\Theta^{a\to b}$. You only say that it is "random noise introduced by the noising process". Also, what happens with this formula if $t^{*}=0$? Is in this case $\hat{X}^{a\to b}$ a function only of this "random noise"?
5. In Definition 3, do you really mean $max$ rather than $sup$? What happens if the domain of $Z$ is unbounded?
6. Item 2 in Assumption 2 suggests that $\hat{G}$ and $G$ have the same dimensionality. But $G$ is some abstract "speaker identity random variable" while $\hat{G}$ is a speaker embedding from some speaker verification system whose dimensionality can be different. What happens if dimensionalities $\hat{G}$ and $G$ do not match?

[2] Diffusion-Based Voice Conversion with Fast Maximum Likelihood Sampling Scheme, Popov et al.

---

### Official Review · Reviewer_YYxq · 2024-11-03

**Soundness:** 1
**Presentation:** 1
**Contribution:** 1
**Rating:** 3
**Confidence:** 3

**Summary:**

The paper tackles the problem of establishing a theoretical framework for understanding how diffusion models can learn disentangled representations. Focusing on the application of voice conversion using non-parallel data, adaptation results on various speech classification tasks are presented.

**Strengths:**

- Interesting theoretical and application area.

**Weaknesses:**

- In general the paper lacks focus, and the interleaving of information-theoretic results regarding disentanglement and details specific to the voice conversion application leads to both heavy notation, and a confusing narrative. The paper's contributions are not clear.
- The experiments are meant to provide validation for the presented theorems, but strong links are not established. Any novel theoretical results may be better served by a separate publication focusing on them. Similarly, to the extent that the experimental methods are able to leverage the theoretical results to make progress, the practical contributions could be made more clear.
- The significance of the experimental results is not established. What is baseline and SOTA performance on these tasks? Furthermore, the methods employed are not described at all in the main body of the paper, and completely relegated to the appendix.
- The paper requires major revision to make both the exposition and the contributions relative to existing work clear.

**Questions:**

See previous section.

---

### Official Review · Reviewer_A4i2 · 2024-11-04

**Soundness:** 3
**Presentation:** 2
**Contribution:** 3
**Rating:** 5
**Confidence:** 3

**Summary:**

This paper focuses on the disentailing (emergent?) abilities of diffusion-models.
It is specifically applied to the speech modeling domain, but its results may be assumed to be more general than the specific domain.
The paper provides some theoretical contributions around disentanglement abilities of the diffusion process, as well as bounds on the "distortions" in case of domain mismatch and domain adaptation.
The paper has a good mix of theory and applied results, and is well written. The theory and specifically the notation is at times hard to follow and complex.

**Strengths:**

The strengths of the paper relate to the good mix between theory and experimental results, much appreciated for a venue as iClear.
Specifically the bounds that relate ensemble improvement rate (EIR)  to the amount of speaker distortion are quite impressive.
The experimental results on different speech-based downstream tasks are solid, as well as the different models comparing against.

**Weaknesses:**

Personally I experienced some difficulties in following the math. Especially the derivations of the different bounds are rather complex, and I am not sure I fully understand the process. One might also comment that - because the basic theory around diffusion models is quite general - perhaps the focus of this paper specifically on speech tasks is rather limiting. It would be interesting to see if these conclusions hold also across modalities, for instance around image based tasks such as detection / segmentation / recognition from disentangled features.

**Questions:**

- if possible simplify notation, especially for equations such at eq9,12 etc - pagination is overflowing
- justify as much as possible assumptions - for instance the second order moments is rather natural and obvious in the derivations - but others such as assumption#2 is less clear what they are needed for / imply
- elaborate on how this work can generalize beyond the speech processing area

---

### Official Review · Reviewer_RTgh · 2024-11-06

**Soundness:** 2
**Presentation:** 1
**Contribution:** 2
**Rating:** 3
**Confidence:** 2

**Summary:**

My understanding of the paper is as follows:
The paper suggests that conditional diffusion models can learn to disentangle features from conditional factors under certain technical conditions. Further, these disentangled features can be useful for downstream tasks on datasets that underwent a distribution shift. However, this disentanglement is not always perfect. So the authors propose a method to mitigate the errors arising in this case by using multiple representations. In this paper they use
The paper presents several definitions, assumptions and theorems related to disentangling independent factors in a generative model.

**Strengths:**

The paper presents some theorems showing that conditional diffusion models can lead to disentangling of representations under certain assumptions. They also show how, using conditional diffusion models for voice cloning, they can create datasets for zero-shot adaptation to train classifiers on these datasets on downstream tasks. By doing different types of aggregation that they call adapt-and-vote scheme, they are able to improve the performance on downstream, zero-shot tasks. Experiments are provided on synthetic tasks that lend support to their theorems. Results are also show on real tasks, with different types of aggregation schemes on predictions from zero-shot datasets constructed to train the downstream tasks.

**Weaknesses:**

Unfortunately, I found the paper quite difficult to parse - either due to my own failings or because the paper is indeed hard to parse. The presentation seems more like a sequence of mathematical observations / proofs strung together into a complicated story, rather than a clear theme. I am happy to change my mind if the authors can help clarify things in the rebuttal process, and I am leaving a large number of questions below to help me understand things better -- with probably more follow up questions if the authors engage.

I think the paper is very poorly motivated in setting up what the authors want to solve, and examples of the exact specifics. It is not clear what the paper is trying to accomplish. Is it a proof that under certain assumptions, a conditional diffusion model can learn somewhat entangled distributions and adapt-and-vote can help in training classifiers on out of distribution data ? In spite of all the proofs, I  am left wondering what the main contribution of the paper is ? Is it adapt-and-vote ? If so, how does it compare to other methods ? It seems that Table 2 and other tables only show comparisons between different types of voice cloning models, using adapt and vote. Is it possible to compare to some other techniques out of distribution ?  What about showing disentangling without voice cloning ?


On the other end of spectrum, related to details in the paper. I found it hard to read because the authors make a lot of implicit assumptions about the terminology and probably wanted to keep the presentation more abstract. However this leads to a paper that's really difficult to parse.  To give some examples - in section 2, is Z a variable that is computed from the data X, or an independent variable ?  Is G computed from X, or just a one-hot label ? These things get resolved reading the paper in more detail, but it takes an inordinate amount of effort to parse, and it leads me to believe there are probably important technical details in the paper that I missed out on, in spite of my best (under a finite budge of time) effort. Please see my questions in the section below.


Finally. About disentangling representations. Generative models using independent variables as inputs has been a guiding principle in representation learning for a long time. It helps make the claim, at test time that new combinations of data can be generated because the variables input into the generative model are independent.  However, it seems to me that this is an artificial setup - for example in the case of style and content, it's really hard to define what is considered content and what is considered style. After learning such models, we often find (and this paper also finds in some cases) (and this is certain true for variational auto encoder like models) that arbitrary combinations of variables that are treated as independent by models have strong dependence on each other. This can result in holes where generating from the prior leads to weird results.  The paper proposes to use mutual information over the population as a proxy for disentanglement, in Definition 2, which I appreciate as a good attempt, but I wonder if such a thing is possible when it come to variables like speakers and content. It is possible that certain speakers have certain topics / content they are more likely to generate. In these cases such a disentanglement may not be possible, or, in some cases even desirable, now that we have really powerful models.   Given these issues, how should I think of $\epsilon$ in $\epsilon$-disentanglement  ? i.e. what does this way of thinking about abstraction of disentanglement lead to ?

**Questions:**

- Line 64 - $X \sim q_{\gamma}$ -- neither $q$, nor $\gamma$ are defined at this point.
- Line 69, 76.  On line 69, $Y$ has been defined as a "downstream classification task Y" in line 69, and then used interchangeably as a label $Y$ on line 71, which can be confusing.  The same sentence refers to variable $Z$ as the "content of the speech..for a downstream classification task $Y$".  Can you please clarify again, what the difference between $Z$, and $Y$ is ? I'm assuming, based on what I see later that Z is a function of X, i.e. $Z(X)$ and this is the input to the diffusion model.
- Line 78 - $d$ is not defined.
- Line 78 Also, why is it important that the mutual information between Z, and G be bounded by this particular quantity ? Is $\epsilon_{\psi}$ an arbitrary quantity ? Then is $\epsilon_{\psi} - \frac{1}{2} log(2\pi e)^d \epsilon_{\psi}$ just another arbitrary constant, and the bound is just over $h(X) + \epsilon$ ? Also it seems that  $\epsilon_{\psi} - \frac{1}{2} log(2\pi e)^d \epsilon_{\psi}$ can be negative or positive -- if so, can you give some further insight on what that implies ?
- Line 89 seems weird.
- On line 92, $\hat{G}_t := g(X_{<t})$ G is a function of all prior noising steps. How do you get this at inference time, when the denoising step actually works in the opposite direction ?
- Line 97 - shows that s is a score function from ($[0,T]$ x $\mathcal{X}$ x $\mathcal{G}$ -> $\mathcal{R}$). Shouldn't the output be $\mathcal{R}^d$ where $d$ is the dimensions of the data ? Also, in equation 4, you should that score function actually receives $z(X)$ so the dimensions should be $[0,T]$ x $\mathcal{Z(\cdot)}$ x $\mathcal{G}$ -> $\mathcal{R}^d$. Please correct me if I am mistaken.
- Line 101 - what does it really mean to only keep the content information of $X_t$ -- this has always been one of the central issues of representation learning where this definition is hand-waved away; what is considered content is quite subjective. Bottleneck representation has always been a vague description.
- Line 121 -- $X^1 \sim q_{\beta,0}$. It is not until later, that its explicitly state that $\beta$ are the parameters of the target distribution.
- Line 124 -- in the denoising direction, the conditioning variable A that is used seems (to me, although its not explicitly stated) to be computed from the source speaker ($X^1$)  ? It seems that G is computed from the second speaker, so I would assume that the auxiliary variables are also computed from the second speaker. Is there a reason why this is structured that way ? The information flow to voice conversion then is both A, and X^T, and if A isn't carefully chosen, it might reveal a lot more to the model.
- Line 134. The definition of implicit bottleneck, seems to be quite undefined (implicit? :)). What is the bottleneck here ? What is the constraint here..?
- Line 164 - $d_{TV}$ is presumably the total variation in distribution, but it's not stated.
- Line 164 - I'm assuming $q_{X^{1->2}|Z^1=z}$ means the generated distribution over the full diffusion model. Is that correct ? With a diffusion model this would be a really strict test because under the same condition z, there are probably a large number of X that satisfy the prior Z=z. Also the distributions is not exactly calculable because the distribution cannot be computed explicitly.
- Line 172 - refers to the "true speaker representation G..". I understand that its know for the synthetic cases like the LSGMM shown in the paper, but is it known in other cases ?
- Line 268: In equation 24 -- is $P_{\alpha}$ supposed to be $P_{\beta}$ ?  I was under the assumption that evaluation would be done over a separate distribution $\beta$.
- Line 258 - Why would one perform speech classification using zero-shot voice-converted speech ? Learning purely on synthetic data can cause models to learn about artifacts of the generative model rather than real data. So I wonder if this would ever work well, if the training data has no speech related to the target distribution. e.g. if the target speech comes from disarthric speech, but training data is normal speech, the generated speech would not what the same distribution as real data, and learning a classifier on this speech would not really learn very meaningful things.

- Line 277 - -I'm afraid I do not follow how to generate random target speaker embedding from a sub-gaussian $\rho$ here. Where did $\rho$ come from ? The model seems to have been trained with conditioning on $G$, which in line 91 was generated from an embedding function on $X_{<t}$.  It seems that $\rho$ doesn't necessarily come from the same distribution. Could you say how it's generated here ? -- Line 277 also, is it related to the target distribution $\beta$ in some way ? It seems that the theorem 4 provides some conditions under which soft voting and hard voting with the multiple individual speaker models lead to bounds on the improvement error rate. Can you say something about what the best way of selecting $\rho$ is ? I would think it should be guided by $\beta$ since that is one way to ensure improvement on the target distribution
-- Line 279 --  What makes the model multi-class ? Since nothing has been provided so far about  what the speech classification task itself is, I'm not sure why this notion about multi-class suddenly got thrown in ? Can you state in clear words what is being trained here ?  It say the loss is $L_{P^G_\alpha}(f)$ -- is the loss computed on $P^G_\alpha$ ? I thought it was computed on the dataset that was just generated, ie., $P^G_\gamma$.
-- In theorem 4, its seems that the ensemble error rate are bounded on the top. Can you say something about minimum improvement from using the hard and soft voting ? I think that's probably more interesting ?

---

### Note · Authors · 2024-11-12

I have read and agree with the venue's withdrawal policy on behalf of myself and my co-authors.